# **Response of Antarctic Ice Sheet Mass Balance to Climate Change**

Jingang Zhan<sup>1</sup>, Hongling Shi<sup>1</sup>, Yong Wang<sup>1</sup>, Yixin Yao<sup>1,2</sup>, Yongbin Wu<sup>3</sup>

<sup>1</sup>State Key Laboratory of Geodesy and Earth's Dynamics, Institute of Geodesy and Geophysics, Chinese Academy of Sciences, Wuhan 430077, China

<sup>2</sup>University of Chinese Academy of Sciences, Beijing 100049, China

<sup>3</sup>Institute of Remote Sensing and Surveying and Mapping of Henan Province, Zhengzhou, 450003, China

Correspondence to: Yong Wang (ywang@whigg.ac.cn)

Abstract. The ice record should have recorded and will likely reflect information on environmental changes such as atmospheric circulation. In this paper, 153 months of Gravity Recovery and Climate Experiment (GRACE) satellite time-

- varying gravity solutions were used to study the principal components of the Antarctic ice sheet mass change and their timefrequency variation. This assessment is based on complex principal component analysis and the wavelet amplitude-period spectrum method to reveal the main climatic factors that affect the change on the ice sheet. The complex principal component analysis results reveal the principal components that affect the mass change of the ice sheet; the wavelet analysis present the time-frequency variation of each component and the possible relationship between each principal component and different
- climatic factors. The results show that the specific climate factors represented by low-frequency signals with a period greater than 5 years dominate the changes of the Antarctic ice sheet mass balance. These climate factors are related to the abnormal sea surface temperature changes in the equatorial Pacific (Niño 1+2 region), the correlation between the low-frequency periodic signal of sea surface temperature anomalies in the equatorial Pacific and the first principal component of the ice sheet mass change in Antarctica is 0.65. The first principal component explains 85.45% of the mass change in the ice sheet. The
- change in the meridional wind at 700 hPa in the South Pacific may be the key factor that determines the effect of sea surface temperature anomalies in the equatorial Pacific on the Antarctic ice sheet. The atmospheric temperature change in Antarctica is the second most important factor that affects the mass balance of the ice sheet in the area, and its contribution to the mass balance of the ice sheet is only 6.35%. This result means that with the increase of low-frequency signals during the El Niño period, Antarctic ice sheet mass changes may intensify.

### 25 1 Introduction

It is well known that Antarctica plays a key role in the earth's climate regulation. The relationship between the ice sheet and climate change over Antarctica has been a recent focus of scientific attention. Studies have shown an elevated rate of atmospheric warming over Antarctica during the second half of the 20th century (Chapman and Walsh, 2007) and strong atmospheric warming over West Antarctica and the western Antarctic Peninsula (WAP) at rates of 0.14°C/decade and 0.402/decade and 2000. The second half of the 2000 for the 2000 Dimensional Content of the 2011 of the 2000 dimensional content of the 2011 of the 2012 dimensional content of the 2011 of the 2011 of the 2012 dimensional content of the 2011 di

0.4°C/decade, respectively (Vaughan et al., 2003; Turner et al., 2005, 2009; Steig et al., 2009; Ding et al., 2011). Data from

air temperature recorders at research stations on the Antarctic Peninsula (AP) also showed this atmospheric warming trend during the second half of the 20th century, especially over the WAP (Turner et al., 2005; Fernandoy et al., 2012). Turner et al. (2016) and Oliva et al. (2017) analyzed stacked temperature records and noted that the atmospheric temperature changes shifted from a warming trend of  $0.32^{\circ}$ C/decade from 1979-1997 to a cooling trend of  $-0.47^{\circ}$ C /decade from 1999-2014. This climate

- change has contributed to the regional mass balance of glaciers. Recently, some researchers (Turner et al., 2016; Clem and Ryan, 2013; Paolo et al., 2018) have also noticed the effect of El Niño changes and atmospheric circulation changes in Antarctica on the mass balance of the AP ice sheet. They pointed out that changes in the mass of the Antarctic ice sheet will increase with the increase in annual atmospheric changes resulting from climate warming.
- Glaciers, and the climate information they contain, play an important role in the study of climate change. For example, Bevan et al. (2017) use a flow-line model and a firn density model (FDM) to date and interpret observations of melt-affected ice layers found within five 90-m boreholes distributed across the ice shelf. They pointed out that units of ice within the boreholes, with densities exceeding those expected under normal dry compaction metamorphism, correspond to two climatic warm periods on the AP within the last 300 years. Some studies (Fernandoy et al. 2012; Laepple et al., 2018) also tried to reconstruct regional ancient climate change or to characterize the present climate situation using ice core data from the northern AP region.
- The ice cores provide a record of climate elements that can be used to interpret changes over time (such as in temperature, precipitation, atmospheric chemistry and atmospheric circulation) and can also provide a record of the various factors that affect climate change (such as greenhouse gases and volcanic activities). However, due to the challenges of working in Antarctica, the number of collected and studied ice cores is low, which limits our understanding of the dynamic relationship between Antarctic ice sheet mass balance and climate change.
- With the development of modern geodetic observation technology, especially the successful launch of the Gravity Recovery and Climate Experiment (GRACE) satellite in 2002, a qualitative leap has been made in monitoring the global gravity field over time. Due to the slow geological tectonic processes deep inside the earth that affect the earth's gravitational field, the cycle of their activity is usually tens of thousands of years. In contrast, the short-term gravitational field change can be seen as a result of changes in surface fluids (Whar et al., 1998). Therefore, the measurement of real-time changes in gravity fields
- using GRACE can be used to monitor local water reserves, glaciers in Greenland, and ice sheet mass changes in Antarctica (Chen et al., 2009, 2011; Xiang et al., 2016; Gardner et al., 2013; Yi et al., 2014). Ice core studies provide data that enable interpretation of historic climates and changes in climate over time, and these interpretations can be used along with GRACE data on the ice mass to help explain the factors affecting changes in the ice sheet mass balance. Compared with traditional ice core data, GRACE satellites can provide real-time data and direct information on ice sheet mass balance changes in Antarctica.
- These measurement data have unique advantages, such as uniform spatial distribution and continuity in observation, which provides a wealth of scientific observations to study the mass balance of the Antarctica ice sheet and its response to climate change. The mass change of the ice sheet in Antarctica is the result of interactions between the atmospheric vapor and the surface water resources; this interaction is closely related to the changes in air humidity, atmospheric temperatures, atmospheric circulation, and other climatic factors in the Antarctic region. The mass change recorded the global and local

climate change information. Based on the ice sheet mass changes, we used the time-varying information and spatial phase changes of its principal components to analyze the climatic factors that recently affected the ice sheet mass balance in the Antarctic.

# 2 Data

# 2.1 GRACE Data

- The variation of earth's gravity field reflects the redistribution of mass inside the earth. Over a short time (compared with geologic time), the variation of the earth's gravity field can be regarded as mass transfer of the earth's surface and shallow fluid. GRACE, which was jointly developed by the U.S. and Germany, has been successfully operating for more than 10 years. Its monthly gravity solutions reflect changes of 1-mm geoid fluctuation at a 300-km spatial scale (Bettadpur et al, 2015; Save et al., 2016) and can be used to monitor gravity field variations caused by changes in hydrology and the cryosphere, earthquakes,
- and glacial isostatic adjustment (Ramillien et al., 2006; Chen et al., 2007; Chen et al., 2008; Velicogna, 2009; Rignot et al., 2011a, 2011b).

The GRACE data used in this paper are the Release-05 (RL05) solutions provided by the Center for Space Research (CSR), University of Texas-Austin. The 153 approximately monthly GRACE gravity solutions cover the period from January 2003 through September 2015 (~12 solutions are missing), each of which consist of normalized spherical harmonic (SH) coefficients,

- to degree and order 60. The main improvements in the new products are the mean model and gravity corrections of various new tidal models. Some processing algorithms and parameters have also been improved, regarding alignments between the star camera data rate, accelerometer, and K-band system (Bettadpur, 2012). Compared with previous data, the RL05 gravity solutions substantially reduce the stripe noise. However, at high degrees and orders, GRACE spherical harmonics are contaminated by noise, including longitudinal stripes, and filtering is still needed. In our study, the smoothness priors method
- (Tarvainen et al., 2002; Zhan. et al., 2015) was used to remove noise in the spatial domain. Compared with the Gaussian filter, correlated-error filter and the combined filter (Gaussian with 300-km smoothing + correlated-error), the smoothness priors method has advantages of less reduction in signal amplitude at high latitude, preservation of greater detail for short-wavelength components in the result, and less signal distortion at low latitude.

### 2.2 Glacial Isostatic Adjustment

- In order to compute ice sheet mass trends from GRACE data and interpret them as changes in the water content of hydrologic basins, ocean bottom pressure, or ice sheet mass, one must remove the effect of the glacial isostatic adjustment (GIA) of the lithosphere and mantle. In this paper, we used the latest ICE-6G\_C model (Peltier W. et al., 2015), which consists of normalized spherical harmonic coefficients to degree and order 256, to remove the effect of GIA. The ICE-6G\_C model has been explicitly refined by applying all of the available global positioning system (GPS) measurements of vertical motion of the crust that may
- constrain the thickness of the local ice cover as well as the timing of its removal. Additional space geodetic constraints have

also been applied to specify the reference frame within which the GPS data are described. The data sets used here will be made available at <u>http://www.atmosp.physics.utoronto.ca/~peltier/</u>.

# 3 Method

## 3.1 Equivalent Water Height

The GRACE monthly gravity solution is usually expressed in the form of normalized spherical harmonic (SH) coefficients. According to Wahr et al. (1998), at an arbitrary point with colatitude  $\theta$  and longitude  $\lambda$ , the ice mass change can be expressed in the form of the surface equivalent water height (EWH) as

$$\Delta\sigma(\theta, \lambda) = \frac{a\rho_e}{3\rho_w} \sum_{n=0}^{\infty} \frac{2n+1}{1+k_n} \sum_{m=0}^n \left\{ \left[ \tilde{c}_n^m \cos(m\lambda) + \tilde{s}_n^m \sin(m\lambda) \right] \tilde{P}_n^m(\cos\theta) \right\}$$
(1)

where  $\rho_e$  is the average density of the earth and  $\rho_w$  is water density. Parameter  $k_n$  and a are respectively the load Love 105 number and the equatorial radius,  $\tilde{P}_n^m(\cos\theta)$  is the *n*th-degree and *m*th-order fully normalized Legendre function. The coefficients  $\tilde{c}_n^m$  and  $\tilde{s}_n^m$  are the normalized SH coefficients.

## 3.2 Complex Principal Component Analysis

Principal component analysis (PCA) is based on the idea of using an orthogonal transformation to convert a set of possibly related variables into a set of linearly uncorrelated variables; these linearly uncorrelated variables are called principal

- components. This transformation is defined according to the guideline that the first principal component has the largest possible variance and therefore accounts for as much of the variability in the data as possible. Each succeeding component in turn has the highest variance possible under the constraint that it is orthogonal to the preceding components. The resulting vectors thus form an uncorrelated orthogonal basis set.
- The disadvantage of the PCA method is that it can only detect standing waves rather than advancing waves due to the absence of phase information. To overcome this, Horel (1984) introduced the phase information into PCA to identify traveling and standing waves (Pfeffer et al., 2010; Kichikawa et al., 2015; Zhan et al., 2017) and named the method complex principal component analysis (CPCA). The CPCA method transforms an original data set and its Hilbert transform into a complex time series and then conducts a principal component analysis by calculating the covariance or complex characteristic vectors of the cross-correlation matrix. This method has been successfully applied to identify traveling wave properties of glacier mass
- changes in the Tibet Plateau. A more detailed explanation can be found in Zhan et al. (2017).

# 3.3 Wavelet Amplitude-period Spectrum Analysis

The Antarctic ice sheet mass balance is under the influence of climate change and exhibits unsteady quasi-periodic change. In fact, these changes in mass balance are the result of variations in time of specific climate factors represented by different frequency signals. Thus, after obtaining the temporal change series of principal components of mass change in Antarctica,

- the time-frequency information from the time series should be analyzed. The wavelet amplitude-period spectrum (Liu, 1999; Liu and Hsu, 2012; Zhan et al., 2003) is a useful tool for analyzing the variation in time of periods and the amplitude (energy) of different signals. Here we chose the Morlet wavelet (Morlet et al., 2012) as the basic wavelet. The advantage of this method is that it is easy to use and we can easily obtaine the time-varying amplitude and period, as well as phase information of each periodic term (or standardized periodic term). This method has been widely used in time-frequency
- analysis of sea level changes, glacier mass balance and other geophysical signal changes. For more detailed description, see

# 4 Mass Change and Its CPCA Analysis

Zhan et al. (2003, 2017), Liu (1999), and Liu and Hsu (2012).

The surface mass change field (in units of equivalent water height) in a regular  $1^{\circ} \times 1^{\circ}$  grid was calculated over the Antarctic using each of the GRACE spherical harmonic solutions following Eq. (1). Then, we filtered each surface mass change field using the smoothness priors method (Tarvainen et al., 2002; Zhan et al., 2015) and interpolated missing data using a spline function at each grid point. The rate of change of the ice sheet mass based on GRACE data was estimated at each grid point using least squares to fit a linear trend, an annual signal and a semi-annual signal. One should note that the  $1^{\circ} \times 1^{\circ}$  gridded data used here do not improve the resolution of GRACE data. The resolution of the calculated data depends on the degree of the RL05 solutions, and the GRACE RL05 solutions are limited by the band-limited nature of GRACE orbit configuration

(inclination, altitude, and separation of the twin satellites), with an approximate resolution of approximately 300 km near the equator (Chen et al., 2017). Relevant information is available from the NASA website (<u>https://grace.jpl.nasa.gov/data/get-data/jpl\_global\_mascons/</u>).

One can also calculate smaller grid data using those solutions, but the smaller calculated grid data do not indicate more short wavelength signals in the results. The accuracy of the calculated data retains 1-mm geoid undulation at an approximate 300-

- 145 km scale. The accuracy of the calculated grid data depends on the accuracy of the RL05 solution itself (Bettadpur et al., 2015; Save et al., 2016), rather than the size of the grid. Figure 1 shows the trend of the ice sheet mass change after eliminating the effect of GIA on the Antarctic from 2003-2015. It can be seen from Figure 1 that the Antarctic ice sheet mass change has the following spatial characteristics: there is a large mass loss area from the AP to Ellsworth (corresponding to the B19-B27 basin). The average annual ice sheet mass loss is 258.5 Gt in this area. There is a small mass drop center in the Ross Land and Victoria
- area (corresponding to the B14-B17 basin), and Wilkes area (corresponding to the B14-B13 basin). The average ice sheet mass loss in these two areas is 28.2 Gt and 15.5 Gt per year. The area of mass increase is mainly located in Maode Queensland and

Enderby Land (corresponding to the B4-B8 basin) areas corresponding to the East Antarctic West Indian Ocean sector. The average ice sheet mass increase in the area is 53.9 Gt per year. The mass of ice sheets in other areas did not change significantly. To analyze the principal components of the Antarctic ice sheet mass change from 2003 to 2015, we used the complex principal 155 component analysis method. Table 1 shows the eigenvalues of the first three principal components of the Antarctic ice sheet mass change and their contribution to the mass change during the period of 2003-2015. As can be seen from Table 1, the eigenvalues corresponding to the first three principal components are 9288.65, 690.32, and 147.22, respectively, and their contributions to the ice sheet mass change in the region are 85.45%, 6.35%, and 1.35%, respectively. The analysis results of the principal components show that the main factors influencing the Antarctic ice sheet mass balance change mainly come 160 from climate change that corresponds to the first principal component. This climate change has absolute dominance over the ice sheet mass changes in the Antarctic region and can explain 85.45% of the area's ice sheet mass changes. Figure 2 shows the time evolution of the principal component, its corresponding spatial mode, and the phase distribution (arrows) of the first two components derived by CPCA. Among them, the ice sheets in the AP and West Antarctica except for Basin 1 and Basin 18, Wilkes land (B13), and Dronning Maud Land and Enderby Land (basins B4-B8) (Fig. 2b) areas are the most sensitive to 165 climate change. The impact of climate change corresponding to the second principal component of the mass change in Antarctica rapidly drops to 6.35%. The ice sheet mass in the area of Ellsworth Land (B20-21) (Fig. 2d) is more sensitive to

climate change corresponding to the second principal component. However, change corresponding to the third principal component only accounts for 1.35% of the mass change in the region. This result shows that the changes in the climatic factors that recently affected the mass balance of the ice sheets in Antarctica are relatively simple.

#### 170 **5 Discussion**

## 5.1 Mass Change in the Antarctic

The Antarctic ice sheet mass change estimated from the GRACE data shows that the mass decline center in the Antarctic area is located in the AP and West Antarctica (except for Basin 1 and Basin 18), Ross and Victoria Land (basins B14-B17), and Wilkes Land (basins B14-B13). Among them, the ice sheet loss in the AP (B24-B27) and the West Antarctic (B19-B23) is the

- most significant, with an average loss of 258.5 Gt/yr. The ice sheet mass loss of Victoria Land and Wilkes Land is relatively slow, with an average loss of 28.2 Gt/yr and 15.5 Gt/yr, respectively. In East Antarctica, only the Dronning Maud Land and Enderby Land (B4-B8) experienced an increase in mass, with an average annual increase of 53.9 Gt. However, even with this mass increase in this region, the overall mass of ice sheets in Antarctica exhibited a declining trend, with an average annual loss of 248.6 Gt. This mass loss translates to a contribution to global sea level rise of 0.65 mm/yr.
- Gardner et al. (2018) used Landsat remote sensing data to analyze the change in ice flow in the Antarctic. The study showed that the ice sheet mass in the West Antarctica (B21, B22, and B20) is in an accelerated state of loss based on changes in ice discharge over different periods, with an average annual loss of 214 Gt. The average annual mass loss of ice sheets in the AP region is 31 Gt. The mass of the East Antarctic ice sheet is in a weak equilibrium, with an average annual increase of 61 Gt.

This result is basically consistent with the results of this paper. Velicogna and Wahr (2006) used the GRACE data from 2002 to 2005 to study changes in the ice sheet mass in the Antarctic region and estimated that the ice sheet mass across Antarctica lost an average of 152 Gt/yr. This result is lower than the results given in this paper. One reason is that Velicogna and Wahr (2006) used an earlier version of GRACE data. Because of the relatively large band noise in earlier versions of GRACE data, a larger Gaussian smoothing radius must be used in later data processing to improve the signal-to-noise ratio; The other reason may be that ice mass loss is simply increasing over time as the climate continues to warm. Jacob et al. (2012) used the mascon method to calculate the impact of global ice sheet and glacial changes on global sea level. They pointed out that the Antarctic

- ice sheet mass lost 165 Gt/yr between 2003 and 2010, with a contribution to sea level rise of  $0.46 \pm 0.2$  mmyr<sup>-1</sup>. Baur et al. (2013) used GRACE data for the nine years from March 2002 to March 2011 to analyze the impact of Antarctic ice sheet mass loss on sea level changes. According to their research results, during the period from March 2002 to March 2011, the Antarctic ice sheet had an increase of 51 Gt/yr in East Antarctica and a loss of 201 Gt/yr in West Antarctica. The contribution to global
- sea level rise was 0.29 mm per year. Luo et al. (2012) used the filter method of combining P3M6 with the 300-km Fan filter to invert the trend of the mass change of the Antarctic ice sheet using the GRACE time-varying gravity solutions from August 2002 to June 2010. They calculated the rate of change of the Antarctic ice sheet annual ablation as 80 Gt. These numbers are lower than the ablation rate of the ice sheet given in this paper. The reason for this difference may be due to two aspects. First, when the stripe noise was removed, the SPM filter method was used to filter the equivalent water column height derived from
- GRACE data in the spatial domain. No filtering was performed on the bit coefficients of the GRACE time-varying gravity solutions during the filtering process. Thus, the amplitude of geophysical signals in the model was preserved to the maximum extent. However, the mascon method and the traditional filtering method performed different degrees of Gaussian smoothing on the bit coefficients of the GRACE model, which weakened the signal amplitude to a certain extent. Zhan et al. (2015) pointed out that compared with the traditional Gaussian and decorrelated error filtering methods, the attenuation of the signal
- by the SPM filtering method can be increased by approximately 17%. Another factor that may cause the differences is that different scholars used different post-glacial rebound models, which is one of the greatest causes of different results. In the Antarctic region, there is still some disagreement about the effect of post-glacial rebound. Differences between different post-glacial rebound models are mainly concentrated in West Antarctica. Although there are numerical differences in the Antarctic ice sheet mass changes given by different scholars, these differences are not sufficient to hide the implications of the change
- in the ice sheet mass in Antarctica. In other words, the Antarctic ice sheet has been losing mass every year in recent years even though some sub-regions gained ice mass. This mass loss reflects, to a certain extent, the Antarctic ice sheet's response to changes in the global climate and environment.

#### 5.2 The Response of the Antarctic Ice Sheet Mass Balance to the Change in the El Niño Low-Frequency Periodic Signal

Figure 3 shows the time-frequency characters of the first two components as well as the sea surface temperature anomaly in the Niño 1+2 region and the air temperature changes over the Antarctic from 2003 to 2015. The wavelet amplitude-period

spectrum of the first principal component time series shows that the principal component contains significant periodic signals

of 8.5 years and 6.5 years (Fig. 3a). The energy of the 8.5-year periodic signal is the largest, occupying the dominant position of mass change in Antarctica, followed by that of the 6.5-year periodic signal and that of the annual periodic signal. That is, the first principal component mainly reflects the change of the low-frequency signal over a period longer than 5 yr. Fig. 3c
shows the wavelet amplitude-period spectrum of the sea surface temperature anomaly time series in the Niño 1+2 region. It can be seen from the figure that the sea surface temperature anomaly in the Niño 1+2 region has signals with periods of 2-3 years in addition to the 8.5-year and 6.5-year periodic signals. The energy of periodic signals above 6.5 years is slightly weaker than the energy of the 2–3-year periodic signals. The lag correlation analysis results show that the correlation coefficient between the first principal component and the sea surface temperature anomaly in the Niño 1+2 region is 0.24 (http://ds.data.jma.go.jp/tcc/tcc/products/elnino/index/). The correlation between the first principal component and the sea surface temperature anomaly is a slight so 0.65, which is much greater than the

0.16 significance level of the 95% confidence based on Monte Carlo hypothesis testing (see Table 2). This result shows that changes in the low-frequency signal of the sea surface temperature anomaly in the Niño1+2 region of the equatorial Pacific Ocean may be the main reason affecting the mass change of the ice sheet in Antarctica. The ice sheet mass in the AP and West

- Antarctica (B19-B27), Wilkes Land (B13), and the Dronning Maud Land and Enderby Land (B4-B8) areas is the most sensitive to the change in the sea surface temperature anomaly in the low-frequency periodic signal in the El Niño phenomenon) with large mass loss in the AP and West Antarctica, Wilkes Land and a small mass increase in the Dronning Maud Land and Enderby Land (Fig. 2b). In West Antarctica, the signal's effect on the ice sheet mass changes is mainly from the eastern South Pacific through the AP into the Ellsworth Land area. In East Antarctica, the effect of this periodic signal on the change in ice
- sheet mass is mainly concentrated in the Dronning Maud Land and Enderby Land areas corresponding to the West Indian Ocean and the Wilkes Land area corresponding to East Indian Ocean.
  Oliva et al. (2017) in analyzing the influence of air temperature changes in the AP on the mass balance of the ice sheet, noted

that temperature changes in the AP during a strong El Niño event had a certain impact on solid precipitation and sea ice accumulation in the region. Their main focus was on solid precipitation, ice sheet ablation, and sea ice changes caused by

- temperature changes, without much analysis of the intrinsic link between the changes in the ice sheet mass in the Antarctic region and the El Niño event. Turner et al. (2016) analyzed the air temperature changes across the AP and found that abnormal changes in the sea surface temperature (El Niño events) in the Eastern Pacific can cause changes in the strength of Rossby waves that propagate southwards, as well as the strengthening of polar front jets, the strengthening of the cold eastward winds in the northern AP, and changes in the accumulation of floating ice. The accumulation of floating ice has an impact on the
- temperature changes in the area. Turner et al. (2016) believed that during the cooling period (1999–2014) of the AP, the subtropical front that spreads to the AP is reflected by the enhanced polar front jet (60°S) and cannot affect the interior of the Antarctic based on the 300-hPa zonal wind component. Obviously, Turner et al. (2016) noticed the effect of the El Niño event on the local atmospheric circulation. Clem and Fogt (2013), by analyzing the relationship between the El Niño-Southern Oscillation (ENSO) and the Southern Annular Mode Index (SAM) over time, found that these climate models have spatial
- dependence with respect to the influence on the AP. There is a clear correlation between ENSO and climate change in the

WAP, and there is a clear correlation between SAM and climate change in the northeastern AP. There is no obvious correlation between temperature changes in this area and the above climate model changes. Paolo et al. (2018) used satellite altimetry data to analyze the relationship between changes in the elevation of ice sheets in the West Antarctica and the local atmospheric circulation driven by El Niño/Southern Oscillation. The analysis indicated that during strong El Niño periods, the height of the

- ice sheet accumulation in the Antarctic Pacific sector is greater than the height of the base melting. However, because the density of the base ice sheet is greater than that of the upper snowpack, the ice sheet elevation increases and the overall mass decreases in West Antarctica corresponding to the Pacific sector. The results of this study reflect the impact of El Niño events on the mass loss of the Antarctic ice sheet. When studying the changes in the mass balance of the AP ice sheet, the above researchers noted the effects of El Niño changes on the atmospheric circulation and temperature changes in the AP and tried
- to determine the effect on the mass changes of the ice sheet in the area. They pointed out that studies correlating ENSO tropical forcing with Pacific sector climate indicators, such as the Amundsen Sea Low strength, sea-ice extent, and AP temperature, found that correlations with ENSO are significant for some seasons but not for others, with reversals of the sign of the correlation from season to season in some cases. The dominant effect of El Niño on the Amundsen Sea ice-shelf mass is the increased basal melting associated with the onshore flow of Circumpolar Deep Water and coastal upwelling as westerly wind
- stress intensifies. On interannual timescales, this basal mass loss anomaly, relative to the longer-term mass loss trend, is partially offset by increased snowfall. This precipitation increase is consistent with the northerly wind anomaly during El Niño events, possibly including increased local moisture uptake from the coastal ocean due to a reduction in regional sea-ice concentration.

The first principal component of the Antarctic ice sheet mass balance change obtained from GRACE data was significantly

- correlated with the low-frequency periodic signal of the sea surface temperature anomaly in the equatorial Pacific region; the correlation coefficient was 0.65. This low-frequency periodic signal plays a key role in the Antarctic ice sheet mass balance. The spatial phase information (Fig. 2b) shows that the main factors influencing the ice sheet mass in the West Antarctica originate from the eastern part of the South Pacific. To analyze the possible impact of the El Niño event on the Antarctic ice sheet, we conducted CPCA analysis on the meridional wind field in the South Pacific region (-80°S--40°S, 120°E-75°W) at
- a height of 700 hPa (<u>ftp://ftp.cdc.noaa.gov/pub/Datasets/ncep.reanalysis/pressure/</u>). The results show that the first two principal components of the meridional wind field in the South Pacific have this low-frequency periodic signal (Fig. 4c-d). The correlation coefficient between this low-frequency signal and the low-frequency signal during the El Niño variation is 0.70. During strong El Niño periods, changes in the meridional wind field at a height of 700 hPa in the South Pacific may affect solid precipitation in Antarctica. This result indicates that the influence of the El Niño change on the Antarctic ice sheet is
- likely to be transmitted through the low-frequency periodic signal of the meridional wind field. The detailed mechanism still needs further study.

#### 5.3 Effect of Temperature Changes on Antarctic Ice Sheet Mass Balance

- The results of the wavelet amplitude-period spectrum of the second principal component time series (Fig. 3b) show that the second principal component that affects the mass balance of the ice sheet in the area includes the strong annual periodic signal and weak signals of the 2-year period. Among them, the energy of the annual periodic signal is the largest, and the energy of long periodic signals of 2 years is very small, and its amplitude is less than 1/10 of the annual periodic signal. The amplitude-period spectrum shows that the specific climate factor represented by annual periodic signal dominates the changes in the second principal component, and its variation is relatively simple. Therefore, the low-frequency signal with a period 2 years has little effect on the change of the second principal component. The annual periodic signal is the main factor that affects this principal component. This result shows that the change in the climatic factors represented by this principal component is relatively simple. The analytical results of the Antarctic Oscillation Index (Antarctic Oscillation, AAO. <a href="http://www.cpc.ncep.noaa.gov/products/precip/CWlink/daily\_ao\_index/history/method.shtml">http://www.cpc.ncep.noaa.gov/products/precip/CWlink/daily\_ao\_index/history/method.shtml</a>.), Southern Annular Mode (SAM, <a href="https://legacy.bas.ac.uk/met/gjma/sam.html">https://legacy.bas.ac.uk/met/gjma/sam.html</a>), and the time series wavelet amplitude-period spectrum of the Antarctic air temperature change show that both the Antarctic Oscillation (Fig. 4a) and the Southern Annular Mode (Fig. 4b) have
- significant annual and 2-year periodic signals. The amplitude of the 2-year periodic signal is comparable to that of the annual periodic signal, indicating that the Antarctic Oscillation Index and the Southern Annular Mode may be greatly affected by the low-frequency signal over a period of 2 years in the change of ENSO. The change in air temperature in the Antarctic region (Fig. 3d) has a stable annual periodic signal and weak half-year periodic characteristics, and the energy of the annual periodic signal has absolute dominance. This energy variation characteristic is similar to the energy distribution of the second principal
- component. Therefore, we infer that the air temperature change in the Antarctic region is the second major factor that affects the change in ice sheet mass in the region. The lag correlation analysis results show that the correlation coefficient between the second principal component and the Antarctic region temperature change is -0.85, and its absolute value is also much larger than the 0.16 significance level of the 95% confidence based on Monte Carlo hypothesis testing. The phase lag of the second principal component is 1 month. A negative correlation coefficient indicates that the two phases are in the opposite
- direction. That is, the abnormal increase in the surface temperature causes a loss of the ice sheet mass, and the abnormal decrease in the surface temperature causes an increase in the ice sheet mass. The air temperature in the Antarctic region degrees decreased bv 0.15 on average per decade over the period 2003-2015 (ftp://ftp.cdc.noaa.gov/pub/Datasets/ncep.reanalysis/). This effect can lead to an overall increase in the mass of the ice sheets in Antarctica, which can explain 6.35% of the change in ice sheet mass in the region. From the second spatial mode (Figure
- 2d), it can be seen that except for the southwestern B21-22 valley, which has a positive value, the other areas have basically all negative values. The spatial phase information shows that the factors affecting the change of the second principal component originate from the Antarctic center, reflecting the influence of the cold high pressure in the Antarctic interior on the Antarctic ice sheet mass balance.

During recent years, several studies have analyzed the evolution of the Antarctic climate during the second half of the 20th 315 century and the beginning of the 21st century; all studies consistently show evidence of the pronounced warming that occurred on and around the AP prior to the 2000s (Vaughan et al., 2003; Turner et al., 2005; Steig et al., 2009; Ding et al., 2011). However, most of these studies, typically using periods spanning between the late 1950s and the early 2000s, often do not include climatic data from the last decade. Moreover, the trends are usually inferred for the whole period of the instrumental series, but the inter-decadal or short-term variability is generally not analyzed.

- Turner et al. (2016) used the stacked and normalized surface air temperature record of the Antarctic Peninsula Observatory to analyze temperature changes in the area. They found that the AP experienced a warming period between 1975 and 1997, with an average temperature increase of 0.32 degrees per decade. During the period from 1999 to 2014, the AP experienced a period of temperature decline, with an average decrease of 0.47 degrees per decade. They further determined that during the cooling period of the AP, the temperature of the sea surface in the eastern Pacific was low and the air temperature on the land was high,
- resulting in the intensification of Rossby waves and polar front jets, and the generation of larger surface cyclones in the WAP. These changes were consistent with the frequent La Niña conditions and the enhancement of the temperature gradient along the meridian. As the enhancement of these atmospheric circulations has promoted the strengthening of the easterly and southeastward winds in the northeastern AP, the sea ice in the eastern waters of AP has accumulated. These circulation changes have also increased the advection of sea ice towards the east coast of the peninsula, which in turn has amplified the decline in
- surface air temperature. Turner et al. (2016) emphasized that decadal temperature changes in the AP are not primarily associated with the drivers of global temperature change but rather reflect the extreme natural internal variability of the regional atmospheric circulation. Oliva et al. (2017) used longer-term temperature data (1950–2015) to analyze the characteristics of temperature changes at different stations in the AP and obtained similar conclusions, i.e., that sea ice is an amplifying factor of the recent cooling and recent regional climate cooling along the AP (1999–2014). It is believed that this cooling trend in the
- AP has slowed the rate of glacial retreat in the region. The period of temperature decline in the AP (1999-2014) indicated in the above-mentioned results coincides with some period of the GRACE time-varying gravity field data from 2003 to 2015 in this paper. The temperature change trend during this period is consistent with the trend of the Antarctic mass change reflected by the change of the second principal component of this article, which reflects the overall increase in the mass of ice sheets in Antarctica.

#### 340 6 Conclusion

During the period from 2003 to 2015, the ice sheet in Antarctica showed a large mass loss, and only some parts of East Antarctica experienced a mild increase in ice sheet mass. The mass loss of the West Antarctic ice sheet plays an important role in the mass loss of the Antarctic ice sheet. The mass loss area is mainly located in the AP and West Antarctica (B19-27) in the Antarctic Pacific sector, with a decrease rate of 258.5 Gt/yr. In the Wilkes Land (B13 basin) in the southeastern Indian Ocean

sector of the Antarctic, the average ice sheet loss is 15.5 Gt/yr, i.e., a state of accelerated loss. The rest of East Antarctica is in

a weakly balanced state, and the overall ice sheet mass has increased moderately, with an average increase of 53.9 Gt/yr. The increase in mass is mainly in the Dronning Maud Land and Enderby Land areas.

The low-frequency signal in the abnormal changes of sea surface temperature in the equatorial Pacific Ocean is the main reason for the recent ice sheet mass change in the Antarctic region. In particular, the low frequency signals with a period greater than

- 5 years in the abnormal changes of the sea surface temperature in the Niño 1+2 region and the first principal component of the Antarctic ice sheet mass change have a strong correlation, with the correlation coefficient reaching 0.65. The effect of the specific climate factors represented by low frequency signals on the Antarctic ice sheet mass balance accounts for 85.45% of the total change in the ice sheet mass in Antarctica. The meridional wind field at 700 hPa over the South Pacific may be the bridge for this signal to affect the Antarctic ice sheet mass balance because the same low frequency signal was found in the
- first and second major components. The detailed influential mechanism still needs further study. The air temperature change is the second major factor that affects the mass balance of the ice sheet in the Antarctic. The temperature change has a strong correlation with the second principal component, with a correlation coefficient of -0.85 and a phase lag of one month. A negative correlation coefficient indicates that the change of the two phases is in the opposite direction, that is, an increase in temperature results in a decrease in the mass of the ice sheet, and a decrease in temperature
- causes an increase in the mass of the ice sheet. The temperature in the Antarctic region has been slowly declining recently, with an average decrease of 0.15 degrees every 10 years. This temperature change has slowed down the loss of ice sheet mass in the Antarctic region to a certain extent, and its contribution to the change in ice sheet mass in the region is 6.35%. There are many factors that affect the mass balance of the ice sheet in Antarctica. We only used the first and second principal

components as examples for analysis in this paper, which explain 91.80% of the ice sheet mass changes in the region. To fully understand the causes of changes in ice sheet mass, other principal components need to be analyzed.

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
