# Peer review of "Response of Antarctic Ice Sheet Mass Balance to Climate Change"

_The Cryosphere, 2018_

## Referee Comment (RC1) · Anonymous Referee #1 · 28 Nov 2018

The paper "Response of Antarctic Ice Sheet Mass Balance to Climate Change" by Zhan et al., deals with an interesting topic about the possible cause of the ice sheet mass changes and to some degree they show quite a interesting connection with the sea surface temperatures in the equatorial regions, i.e. global-scale climate change. In order to do that the authors employ the complex principal component analysis that includes the information about the phase, which I think is the key element in this new analysis.

However this paper is "incomplete". The title ambitiously claims to discuss the ice sheet changes in relation to climate changes, but the paper does not provide strong evidence on this. And the presentation is in many ways lacking and the authors dilute too much the actual new information with a lot of redundant and/or consolidated knowledge that

can be easily and abundantly found in the literature.

Part of the main analysis and related results looks interesting and original to me and it is worth publishing, but with substantial major revision.

The CPCA is a good idea to investigate not only "stationary" signal but propagation of disturbances. The method however has some potential drawbacks (mostly related at the not straightforward interpretation of the results, even worse than with conventional PCA because both the amplitude and phase relationships need to be considered), and no element about the reliability of this analysis is provided here. In my understanding the principal component extracted by the authors is different from the conventional principal component analysis (PCA), allowing in principle to identify also propagating signals, and therefore investigate both the space and time behavior. Conventional PCA have been employed in Antarctica in the past mostly to reveal the trends only. I believe that the additional information that Complex PCA can provide about the phase changes, can improve the power of analysis of the GRACE signal, and it would be very nice to see a discussion of the new insight that Complex PCA can provide in this specific case over Conventional PCA. However, and unfortunately, the authors do not show that.

In addition, the analysis of the other climate variables, like El Nino, ENSO, wind and others, is shown in terms of wavelets analysis and the correlation only, that is by its nature only a partial information, and too late in the discussion, while it should be done earlier (in the result session) and the principal component for those variable should be shown too (at least the temporal part). The claimed response of the ice sheet to the climate changes is only supported by correlation index between the climate forcings analyzed and the first 2 principal components of the GRACE-derived signal. The author themselves recognize that this is not a proof, and the last line of the conclusion is "To fully understand the causes of changes in ice sheet mass, other principal components need to be analyzed".

[Figure]

I believe, that analyzing more component is not the point, but rather taking into account other possible phenomena as forcing of the ice sheet changes. Therefore the claimed implication "climate forcing -> mass variation" is too strong and it is not supported. And considering that the Complex PCA are not common knowledge at least for some of the potentially interested readers, there should be more explanations on how to read the data, which are not easy understandable for everyone, otherwise. Referring to a previous paper is not helping to make the present paper more readable.

Another very weak point of this paper is that the analysis of the GRACE data leads the authors to unreasonable estimates of the mass balance in Antarctica. Not enough details are given to be able to point out what could be the reason for this, but the results are completely outside the realm of the possibility, given the strong consensus emerged in the recent years in the scientific community concerning GRACE (and not only GRACE) mass balance in Antarctica (see Shepherd et al. 2012 and 2018 for example).

The authors devote far too much space to discussing this mass balance (it is also put as the first item in the conclusion), comparing with works that are outdated, and not discussing any of the actual relevant work on the topic. A weak explanation about their results being quite "different" from the main literature just demote the whole work. Most importantly, it can be true that the CPCA analysis that the author perform on the GRACE data is not affected by this, but the fact that the GRACE-derived mass balance is so "off", raises the doubt that the processing could be correct, but the data to process could be wrong.

In the following I make more detailed comments.

L39-49: Too much. Shorten or remove it. It's not relevant for the paper.

L52-55: Here GIA, which is not a tectonic process, is missing from the picture.

L62-65: Here the Ocean interaction (which is the most important) is missing from the

picture.

L77: Why RL05 and not RL06? Since there's a new release, the reason of choice of the release should be mentioned.

L80-88: Since the RL06, the description of improvement in RL05 is outdated. I can understand the study has been performed on "old" release, but more words should be spent also on the new release RL06, so not to give the idea that RL05 is the last one or the most up to date.

L92: ICE-6G is inadequate for Antarctica. IJ05-r2 or W12 would be a much better choice. Since 2012 there are several papers about GIA in Antarctica, and several GIA models are available (Caron et al 2018, is one of the most recent and available). I understand that for this work trends like GIA do not matter, but I still don't understand why chosing one of the most inadequate GIA for Antarctica. Probably not using GIA would be a better option, and it would be a good idea to make a comparison with and without GIA correction: to show that makes no difference.

L100: Section 3.1 is useless, we already know all this. Use only formulas that are new or serve some purpose in the paper.

L109/L113: I think that the correct terminology is "linearly independent variables" not "uncorrelated", which has a different mathematical/statistical meaning.

L135: I don't understand how there could be missing points? Are the missing point in time?

L145: The accuracy of RL05 is lower than RL06. A reader might wonder again why not using RL06.

L147: The GIA component is the most inaccurate. Did you take that into account? How? With only one GIA model without errors as the ICE-6G is impossible to have the correct idea of the uncertainty on GIA.

L147-149: All this stuff if it were correct would be redundand with all the previous work, so it's rather useless to discuss it here. It should at least put in comparison with previous works and with the ESA CCI AMB for example, which is based on published work (http://esa-icesheets-antarctica-cci.org/index.php?q=GMB and https://data1.geo.tu-dresden.de/ais_gmb/).

The pattern found by the authors is quite the same as fig.1 (from the esa cci amb) but the ampliture are not. B19-B27 mass balance should be about -180 Gt/yr and not 258.5 Gt/yr:

AIS19 basin 19 $0.7 \pm 3.1$

AIS20 basin 20 $-35.9 \pm 5.6$

AIS21 basin 21 $-55.0 \pm 8.1$

AIS22 basin 22 $-50.8 \pm 9.8$

AIS23 basin 23 $-9.4 \pm 4.9$

AIS24 basin 24 $-10.7 \pm 4.7$

AIS27 basin 27 $1.0 \pm 3.5$

AIS28 NAP(b25-26) $-18.9 \pm 5.9$

TOT $-179.0 \pm 17.2$

L160-169: In this paragraph the use of esperssion "climate change" gives the strong impression that the authors have already drawn their conclusion. But they have yet to demonstrate a clear relationship between climate changes and the ice sheet changes they extract in their analysis. So here and in the rest of the paragraph, I wouldn't call it "climate change" of the fist component. Here just call it "the behavior" of the first component.

L166: I'd like to see also the 3rd component. (instead of figure 1, which is useless)
L167-168: So far, the authors have not shown a correlation of the climate change with the behavior of the principal component, so that statement is not true.

L172-179: Totally redundant (with text above) and wrong.

L184: This work should be compared with GRACE based work (not with altimetry-based work). Altimetry and other techniques have much lower accuracy when it comes to mass trend. The most robust results obtained with GRACE are not in agreement with the mass changes obtained here. See also Shepherd et al. 2018. So the statement is false.

L184: Velicogna and Wahr (2006)?? This is absolutely outdated. They used a very early release of GRACE, a too short time series and a totally unsuitable ensamble of GIA models. It shouldn't be used as reference, for studies that use later release and much longer time series.

L184-189: Totally useless discussion (see also my reason in the above introduction).

L197-198: Exactly! And you didn't mention the most relevant papers anyway.

L198: "this difference may be due...". Since Shepherd et al 2012 it is clear that all the GRACE derived data agree very well when all the input ingredients are the same and even if the methods are different. So no more excuses. If the numbers doesn't match it means that the authors used something quite different in their processing and it's most likely wrong. GRACE derived estimated are well consolidated, so at this point there's no much room left for discussion here. So there is only one thing to do, find the error and fix it.

L208-210: I agree on this but it's not a good excuse for getting the numbers wrong.

L212: And here again, the authors put their conclusion before showing solid prof of that. As I said before at this point the authors hve not shown the correlation with the global climate change pattern yet.

L213: Section 5.2, This part is important and actually show some correlation with the principal component and the global climate dynamic. This must be moved before the discussion as part of the result! Before claiming that the principal component follows a climate change pattern. Note that I am expert in ice sheet changes and not strictly an expert in global climatology, so this section 5.2 is rather difficult to read for me (and more than half of your readers would be like me). So I will point out where it would be nice to have little more information in the text rather than go on internet digging them out from the literature. Here I also note that the previous section about ice mass changes were exaggerately detailed (and I criticised that), if a reader is not expert in ice sheet changes I admit that some more details could be useful, yet it should be done taking the state of the art into account correctly, which has not been done by the authors.

L219: ... rephrase please. Maybe you mean: reflects changes in the low frequency with 5 year or longer periodicity?

L222: The 8.5 and 6.5 years periodic signal in El Nino makes an indeed interesting correlation, but it's not a proof. Since it's not the main energy, do the authors know where this energy is coming from? What are the phenomena that affects that periodicity in El Nino? Is is possible that it is actually a feedback from the ice sheet itself?

L223: "Lag correlation analysis", what does it mean?

L225: what the data in the link are meant to be used by the reader?

L226-227: "...which is much greater..." I understand the meaning but this can be rephrased better.

L228-229: This is interesting! Not robust at all but very interesting. And could it be the opposite, i.e. the ice changes affecting El Nino? I remember previous studies (mostly posters) about this in the early stages of GRACE, but they have never been published because (with short time series) the correlation was weak. Here it's clearly

more visible, but it is not presented in a very convincing way. The study and especially the presentation can be improved.

L230-236: Now you can say this (not before as you did prematurely).

L259: "researchers" -> studies

L269/L271: The are at least two "was" that should be "is".

L277: which signals are you referring to?

L282: Section 5.3. This discussion is about the second component, which has annual periodicity as many other phenomena. It's rather meaningless to find a correlation with an annual signal when you could find a good correlation with any other annual signal in the world. So which one is the most relevant in this case? I think a correlation analysis is insufficient here to determine that. So this is much less interesting and even less solid than the discussion about the first component. L300: For the same reason above, the inference here is extremely weak.

L337-339: This sentence is weird and if the meaning is correct it is really weak. The second component by definition cannot reflect "the overall increase in the mass of ice sheets in Antarctica". Overall increase that is not happening by the way.

L342-347: Same as above. The GRACE derived mass loss are too strong. And even if they were right they are not worth mentioning in the conclusion, since the mass balance is not the aim of the paper. They could be barely used as validation tool... validation that failed in this case.

L356: Could the correlation of annual signal be due to other factors? As I said, there are many phenomena in the world with annual periodicity, each of them would correlate very well or even better. So this is not a solid result.

L365: Rather than other components I'd say that other phenomena should be included in the analysis and eventually excluded from the list of possible cause (or effects).

The figures are of very poor resolution and not easy to read.

[Figure]

[Figure]

**Fig. 1.**

---

## Referee Comment (RC2) · Anonymous Referee #2 · 5 Dec 2018

General Comments: This paper uses complex principal component analysis and wavelet amplitude-period spectrum analysis to examine the main drivers of Antarctic mass change. While the study of the effect of quasi-periodic climate patterns such as the El Nino on Antarctica are important, it is misleading to mostly attribute Antarctic mass balance to such events. Furthermore, the GRACE analysis in the paper suffers from some inadequacies. These issues are explained below. Overall, the paper in its current state is not suitable for publication as it provides misleading conclusions. Significant major revisions are required before potential further review.

The paper assumes that Antarctic changes are caused by low-frequency quasi-periodic climate phenomena and atmospheric circulation patterns, and attempts to divide the attribution of ice sheet mass balance change to such events. However, no evidence is

provided for this assumption and yet this is a major underlying assumption of the study. This is misleading, as these events play a much smaller role in the mass balance of the AIS compared to non-cyclic long term patterns such as enhanced ice-ocean interaction and ice discharge, intrusion of warm saline water on retrograde slopes, etc. The assumption of this paper can lead to misleading conclusions as the oscillatory climate events play a much smaller role in the recent mass balance. Yet the authors attribute the components of the CPCA to climate change as far as its effects on periodic climate phenomena such as the El Nino. For example the authors claim "This result shows that changes in the low-frequency signal of the sea surface temperature anomaly in the Niño1+2 region of the equatorial Pacific Ocean may be the main reason affecting the mass change of the ice sheet in Antarctica." Again it is misleading to attribute cyclic SST anomalies from El Nino to the main driver of AIS mass balance change. Also one has to be careful about correlations. The authors admit that the mechanisms of this proposed relationship have to be explored further in future studies, but one has the be careful with correlations between 5+ year low-frequency SST changes associated with the El Nino and Antarctic mass balance. The conclusion that air temperature is the second dominant effect on the mass of AIS is also based on the correlation between the components of quasi-periodic atmospheric circulation patterns and AIS mass balance, and the same issues and assumptions arise here. The claims of the paper should not be generalized outside of the scope of the study.

As noted in the paper, the 1x1 grid does not represent the true GRACE resolution. Given that the mass change is obtained by simply fitting the time-series for each grid on a smoothed field, it must be noted that the grids are spatially correlated and the trends of nearby basins (particularly small basins) cannot be considered separately from each other. Furthermore, there is amplitude loss in spatial smoothing so the smoothed spatial field is not the optimal way of getting regional estimates. The authors should use a synthetic field to justify their results (compare true vs. retrieved signal) or alternatively use a mascon solution, which is linked in the paper but never used. The overall loss trend of 248.6 Gt/yr seems really high with respect to other estimates.

The only agreement was with a selected altimetry estimate. The authors claim the discrepancies with previous studies such as Velicogna and Wahr (2006) are partly due to previous releases of GRACE data and signal attenuation due to smoothing. However, more recent results are also in disagreement (such as Velicogna et al 2014), which use newer releases and a mascon approach. Furthermore, scaling factors were calculated for previous studies using synthetic fields to account for signal attenuation. And such approaches such as the spherical cap approach are in close agreement with other mascon solutions such as the JPL or CSR mascons. The authors should also consider the official mascon solutions of the processing centers as a point of comparison. While previous studies that are claimed here to be suffering from signal attenuation due to smoothing looked at scaling and gain factors, this study does not make an attempt on quantifying any attenuation with a synthetic field.

Analysis is up to September 2015. Is there a reason for this? Even with the presence of accelerometer issues near the end of the mission, at least another year can be added.

Specific Comments: Line 62: "The mass change of the ice sheet in Antarctica is the result of interactions between the atmospheric vapor and the surface water resources;" this is a misleading statement. While ultimately the mass balance is the result of surface mass balance (including precipitation) and discharge, this is the interaction of many factors such as ice-ocean interaction, bathymetry, etc. Needs to be clarified. Line 72: Mission ended after 15 years. Equation 1: If solving for surface density on the left hand side, the coefficient should not be divided by the density of water. Refer to equation (14) of Wahr et al (1998). The coefficient also needs to be in surface density units. Lines 121-125: this assumes all changes in Antarctica are caused by periodic climate events (such as El Nino etc.). This is not necessarily true and is unjustified. There could well be significant long-term non-periodic changes that are the main drivers of change.

Technical Comments: Line 64: sentence is not very clear. And careful with tense. Maybe "the mass change record contains global and local climate change information

across time".

Line 115: delete "the" in front of phase information.

Lines 147,150: change to "basins B19 to B27", and the corresponding numbers in line 150.

Line 237: "in analyzing the influence of..." makes the sentence longwinded and hard to follow. I think that segment can be removed.

Line 238: "had a certain impact" is very vague. You need to be more clear as to their conclusion.

Line 272: Delete "the" before West Antarctica.

Line 345: "i.e." is not appropriate here as the acceleration does not follow from the trend magnitude, it is a separate fact (and it would be helpful to also report the acceleration value if this is the case).
* * *

---

## Author Comment (AC1) · 3 Jan 2019

➢ However this paper is "incomplete". The title ambitiously claims to discuss the ice sheet changes in relation to climate changes, but the paper does not provide strong evidence on this. And the presentation is in many ways lacking and the authors dilute too much the actual new information with a lot of redundant and/or consolidated knowledge that can be easily and abundantly found in the literature. Part of the main analysis and related results looks interesting and original to me and it is worth publishing, but with substantial major revision.

**Response:**

Based on your suggestion, we have supplemented some content to complete the manuscript. Supplemented content include: (1) Selected literature, words and formulas in the Methods section to illustrate the reliability of CPCA, and additional detail was added in the Discussion section to increase the interpretation of the results and increase reader comprehension. (2) We analyzed the climate variables in the early part of the Discussion and supplemented the related principal component for those variables, such as El Niño, ENSO and wind time series. (3) We revised the last line of the conclusion sentence to "To fully understand the causes of changes in ice sheet mass, other phenomena such as enhanced ice-ocean interaction and ice discharge, intrusion of warm saline water on retrograde slopes should be included in the analysis and eventually excluded from the list of possible effects" to make a clearer statement based on the suggestion. (4) The related sentences were also revised and the theoretical formulas of CPCA

and more explanation of how to read the data were also supplemented in the Methods and Discussion sections. (5) We used the mask method with the drainage basins boundary data definitions by Zwally et al. (2012) to avoid the signal leakage effect in estimation of the mass balance. (6) Related literature is also supplemented in the reference section.

Revised include:

**Pg4-5 lines 109-138:**

"Before CPCA, a complex observation sequence should first be constructed using a real observation series. For a time varying observation vector $u_j(t)$, its Fourier expansion is:

$$u_j(\text{t}) = \sum_\omega \left[ a_j(\omega)\cos(\omega t) + b_j(\omega)\sin(\omega t) \right]. \tag{1}$$

In the above expansion, $j$ stands for the location of the observation point, ωis the Fourier frequency and $t$ is the observation time. In order to describe the propagation characteristics of a time series, it is necessary to construct the imaginary part and convert it into a complex observation sequence. The complex observation sequence can be expressed as:

$$U_j(t) = \sum_\omega c_j(\omega)e^{-i\omega t}. \tag{2}$$

Here, we define $c_j(\omega) = a_j(\omega) + ib_j(\omega), \text{i} = \sqrt{-1}$. Then the Eq. (2) can be expanded as:

$$\begin{aligned} U_j(t) &= \sum_\omega \left[ a_j(\omega)\cos(\omega t) + b_j(\omega)\sin(\omega t) \right] + i\left[ b_j(\omega)\cos(\omega t) - a_j(\omega)\sin(\omega t) \right] \\ &= u_j(t) + i\text{v}_j(t) \end{aligned} \tag{3}$$

The real part of Eq. (3) is the original observation sequence and the imaginary part is the Hilbert transform of the real part, which does not change the amplitude of each component of $u_j(t)$. However, the phase of each spectral component is advanced by $\pi/2$.

The traditional PCA is the principal component analysis of the real observation vector, whereas the CPCA analysis is the principal component analysis of the complex vector constructed. After the normalization of the complex observation vectors, that is the average value is subtracted from the complex observation vector of each observation point, and then divided by the standard deviation

the complex correlation matrix of the observation point can be expressed as:

$$
\begin{bmatrix}
r_{11} & r_{12} & \cdots & r_{1n} \\
r_{21} & r_{22} & \cdots & r_{2n} \\
\vdots & \cdots & \cdots & \vdots \\
r_{n1} & r_{n2} & \cdots & r_{nn}
\end{bmatrix}.
\tag{4}
$$

Here $r_{jk}$ represents the multiple correlation coefficients between the $j$th and $k$th observation points.

CPCA compresses information using the least complex eigenvector $e_{jn}$ of correlation matrix (Eq. 4) and the complex principal component $p_n(t)$, because the correlation matrix (Eq.4) is a Hermitian matrix including n real eigenvalues $\lambda$. $\lambda_j / \sum_{i=1}^{n} \lambda_i$ denotes the contribution percentage of the $j$th principal component.

Observation vector $U_j(t)$ can be expressed as the sum of N principal components,

$$
U_j(t) = \sum_{n=1}^{N} e_{jn}^{*} p_n(t),
\tag{5}
$$

where * stands for the complex conjugate, and both complex principal components and complex eigenvectors are orthogonal. The $n$th complex eigenvector element $e_{jn}$ can be expressed as

$$
e_{jn} = \left[ U_j(t) * p_n(t) \right]_t = s_{jn} e^{i\theta_{jn}} .
\tag{6}
$$

Where, $e_{jn}$ indicates the multiple correlation relationship between the jth time sequence and nth principal component. $s_{jn}$ and $\theta_{jn}$ are respectively correlative order of magnitude and phase. $\left[ \cdots \right]_t$ signifies the average of times. The time sequence elements of principal components can be expressed as the functional form of amplitude $T_n$ and phase $\Phi_n$.

$$
P_n(t) = T_n(t) e^{i\Phi_n(t)} .
\tag{7}
$$
”.

**Pg8 line 218:**

"The Response of the Antarctic Ice Sheet Mass Balance to the behavior corresponding the first principal component"

**Pg8 lines 219-226:**

"Figure 2 shows the time evolution of the principal component, its corresponding spatial mode, and the phase distribution (arrows) of the first three components derived by CPCA. The spatial mode shows where the mass balance is the most sensitive to the change of its corresponding principal component, the phase distribution indicates the source direction of the possible factors that affected mass balance and the length of the arrow reflects the extent to which the mass in this region responds to the variation of these possible factors. From the phase distribution of first principal component (Fig 2b.), we can see that the factors affecting the mass balance mainly come from the direction of the eastern South Pacific. The ice sheets in the AP and West Antarctica (basins B20 to B27), Wilkes Land (basin B13), and Dronning Maud Land and Enderby Land (basins B4 to B8) (Fig. 2b) areas are the most sensitive to the first principal component change."

**Pg9 lines 240-260:**

"The Antarctic Oscillation Index (Antarctic Oscillation, AAO. http://www.cpc.ncep.noaa.gov/products/precip/CWlink/daily_ao_index/history/method.shtml .), Southern Annular Mode (SAM, https://legacy.bas.ac.uk/met/gjma/sam.html), air temperature in the Antarctic region (ftp://ftp.cdc.noaa.gov/pub/Datasets/ncep.reanalysis/), and the meridian wind speed in the South Pacific region ($-80°$ S$-$ $-40°$S) at a height of 700 hPa (ftp://ftp.cdc.noaa.gov/pub/Datasets/ncep.reanalysis/pressure/) were also analyzed using wavelet amplitude-period spectrum method to study the possible relations of this low-frequency signal between different data set. The results of their wavelet amplitude-period spectrum are presented in Fig. 4 and Fig. 5. Both the Antarctic Oscillation (Fig. 4d) and the Southern Annular Mode (Fig. 4f) have significant annual and 2-year periodic signals, the amplitude of the 2-year periodic signal is comparable to that of the annual periodic signal. The Antarctic Oscillation Index and the Southern Annular Mode may be greatly affected by the

low-frequency signal during a period during 2 years in the change of ENSO and have a smaller correlation with the first component of ice sheet. However, the results of the first two principal component of meridian wind speed (Fig. 5a and 5b) show that the meridian wind in the South Pacific region has 8.5-year and 6.5-year periodic signals. The correlation coefficient of the low-frequency signal between the meridional wind field and sea surface temperature anomaly in the equatorial Pacific is 0.77. These results indicate that changes in the low-frequency signal of the sea surface temperature anomaly in the Niño1+2 region of the equatorial Pacific Ocean may be the possible factors affecting the mass change of the ice sheet in Antarctica. The phase distribution information (arrows in Fig. 2b) also indicates that the factors affecting the mass balance mainly come from the direction of the eastern South Pacific, and it is more likely that the sea surface temperature anomaly causes changes in atmospheric pressure and meridional wind, and conducts its effect (such as changes in atmospheric circulation, precipitation, enhanced ice-ocean interaction, ice discharge, intrusion of warm saline water on retrograde slopes and etc.) to the Antarctic ice sheet, because the change of the first principal component of ice sheet lags behind that of the low-frequency signal of the sea surface temperature anomaly by a month (Table 2).

**Pg10 line 291:**

"5.3 Effect of the second and third principal components on Antarctic Ice Sheet Mass Balance"

**Pg10-11 lines 299-331:**

"The phase distribution information (arrows in Fig. 2d) indicates that the factors affecting the mass balance mainly come from the South Pole. This allows us to relate to the temperature changes, as the Antarctic Central area is the source of cold and high pressure air in the region. The wavelet amplitude-period spectrum (Fig. 4h) also shows that the air temperature in the Antarctic region has similar periodic signals.

Figure 2f shows the spatial mode and phase distribution (arrows) of the third components. The phase distribution of the third principal component shows that the factors affecting the mass balance are mainly along the latitude line. The ice sheets in the basins B21−23 and basin B1 areas are the most sensitive to third principal component change. The wavelet amplitude-period spectrum of the

third principal component time series (Fig. 3c) shows that the principal component contains significant periodic signals of 8.5 years, 4 years and 5 years. The energy of the 8.5-year periodic signal is the largest, followed by that of the 4-year periodic signal and the 5-year periodic signal, the energy of the signals with period below 2 years is unstable. From the perspective of phase distribution and cycle components, these factors that affect the third principal component may be correlated with the Southern Annular model index ( Fig. 4b and Fig. 4f) as well as the sea surface temperature anomaly in the Niño 1+2 region during this period. The direction of phase in the basins B21−23 is counterclockwise, while in the basin B18 and the northern of basin B1, the phase is in a clockwise direction. This data suggests that the impact factors may come from the disturbance of small scale local atmospheric circulation.

Raphael et al. (2016) studied the Amundsen Sea Low (ASL) and found that the Amundsen–Bellingshausen Sea (ABS) region exhibits large inter-annual atmospheric circulation variability. This is due, in part, to orographic forcing and in part to its location in the South Pacific, where atmospheric Rossby waves associated with ENSO variability have a year-round influence. The ENSO plays a significant role in determining the depth of the ASL. The most energetic Rossby waves associated with ENSO variability in the Southern Hemisphere occur in spring, and hence the strongest correlations between ENSO variability and the ASL generally occur in this season. In its La Niña phase, in spring, ENSO is associated with a deeper ASL and with warm air advection toward the Antarctic Peninsula and West Antarctica. However, from spring to summer the sign of the correlation of the phase of ENSO with respect to air temperature anomalies over Antarctica reverses in many locations. The ASL is an important circulation feature that influences West Antarctic climate variability. Observations reveal that the ASL has deepened in recent decades with potential impacts on the regional climate through its influence on the meridional wind field. Some studies have suggested that tropical teleconnections have contributed to atmospheric warming in West Antarctica and across the peninsula (Ding et al. 2011; Schneider et al. 2012), and to sea ice loss in the Bellingshausen Sea (Li et al. 2014). The ASL is probably related to the variability of the SAM (e.g., Fogt et al. 2011) and ENSO (e.g., Lachlan-Cope and Connolley 2006). Paolo et al. (2018) noted out that studies correlating ENSO tropical forcing with Pacific sector climate indicators, such as the Amundsen Sea Low strength, sea-ice extent, and AP temperature, found that correlations with

ENSO are significant for some seasons but not for others, with reversals of the sign of the correlation from season to season in some cases. The dominant effect of El Niño on the Amundsen Sea ice-shelf mass is the increased basal melting associated with the onshore flow of Circumpolar Deep Water and coastal upwelling as westerly wind stress intensifies. "

**Pg12 lines 353-365:**

"The effect of the specific factor represented by annual periodicity signals on the Antarctic ice sheet mass balance accounts for 2.57% of the total change in the ice sheet mass in Antarctica. The effect of the third component, which contains significant periodic signals of 8.5 years and 4-5 years, on the Antarctic ice sheet mass balance accounts for 1.87% of the total change in the ice sheet mass in Antarctica. The factors represented by the third component may be related to the small scale local atmospheric circulation change Southern Annular model index and westerly wind from the periodicity of signals and phase distribution.

There are many factors that affect the mass balance of the ice sheet in Antarctica. In addition to the factors of long term ocean driven, we also found that the low-frequency quasi-periodic signals appears in the first principal component based on GRACE data during 2003-2016, which is also maybe another important factors affected the mass balance in Antarctic. To fully understand the causes of changes in ice sheet mass, other phenomena such as enhanced ice-ocean interaction and ice discharge, intrusion of warm saline water on retrograde slopes should be included in the analysis and eventually excluded from the list of possible effects."

**The related reference were also supplemented.**

(1) Rignot, E. & Thomas, R. H. Mass balance of polar ice sheets. Science 297,1502–1506, 2002.

(2) Wingham, D. J., Ridout, A. J., Scharroo, R., Arthern, R. J. & Shum, C. K. Antarctic elevation change from 1992 to 1996. Science 282, 456–458, 1998.

(3) Velicogna, I. & Wahr, J. Measurements of time-variable gravity show mass loss in Antarctica. Science 311, 1754–1756, 2006.

(4) van Wessem, J. M. et al. Modelling the climate and surface mass balance of polar ice sheets using RACMO2–part 2: Antarctica (1979–2016). Cryosphere 12, 1479–1498 (2018).

(5) Shepherd, A., Ivins, E., Rignot, E., Smith, B., van den Broeke, M., Velicogna, I., Whitehouse,

P., Briggs, K., Joughin, I., Krinner, G., et al.: Mass balance of the Antarctic Ice Sheet from 1992 to 2017, Nature, 556, 219–222, doi:10.1038/s41586-018-0179-y, 2018.

(6) Harlander, U., Larcher, T., Wright, G. B., Hoff, M., Alexandrov, K. & C. Egbers: Orthogonal decomposition methods to analyze PIV, LDV, and thermography data of thermally driven rotating annulus laboratory experiments. In Modeling Atmospheric and Oceanic Flows (ed. Larcher, T. & Williams, P. D.), 315–336, 2014.

(7) Hoff, M., Harlander, U., & C. Egbers: Experimental survey of linear and nonlinear inertial waves and wave instabilities in a spherical shell. Journal of Fluid Mechanics, 789, 589-616, 2016.

(8) Von Larcher, T., Viazzo, S., Harlander, U., Vincze, M., & A. Randriamampianina: Instabilities and small-scale waves within the Stewartson layers of a thermally driven rotating annulus. Journal of Fluid Mechanics, 841, 380-407, 2018.

(9) Walcker, R., E.J. Anthony, C. Cassou, R.C. Aller, A. Gardel, C. Proisy, J-M Martinez, F. Fromard: Fluctuations in the extent of mangroves driven by multi-decadal changes in North Atlantic waves. Journal of Biogeography, 42, 2209-2219, 2015.

(10) Nagler;T., D. Floricioiu, A. Groh, M. Horwath, A. Kusk, A. Muir, J. Wuite. Algorithm Theoretical Basis Document (ATBD) for the Antarctic_Ice_Sheet_cci project of ESA's Climate Change Initiative, version 2.1, 15 December 2017.

(11) Raphael, M.N., et al., 2016. The Amundsen Sea low: variability, change, and impact onAntarctic climate. B Am Meteorol Soc., 97, 111–121.

(12) Schneider, D. P., C. Deser, and Y. Okumura: An assessment and interpretation of the observed warming of West Antarctica in the austral spring. Climate Dyn., 38, 323–347, 2012.

(13) Li, X., D. M. Holland, E. P. Gerber, and C. Yoo: Impacts of the north and tropical Atlantic Ocean on the Antarctic Peninsula and sea ice. Nature, 505, 538–542, 2014.

(14) Fogt, R. L., D. H. Bromwich, and K. M. Hines: Understanding the SAM influence on the South Pacific ENSO teleconnection. Climate Dyn.,36, 1555–1576, 2011.

(15) Lachlan-Cope, T. A., and W. M. Connolley: Teleconnections between the tropical Pacific and the Amundsen-Bellingshausen Sea: Role of the El Niño/Southern Oscillation. J. Geophys. Res., 111, D23101, 2006.

➤ The CPCA is a good idea to investigate not only "stationary" signal but propagation of disturbances. The method however has some potential drawbacks (mostly related at the not straightforward interpretation of the results, even worse than with conventional PCA because both the amplitude and phase relationships need to be considered), and no element about the reliability of this analysis is provided here. In my understanding the principal component extracted by the authors is different from the conventional principal component analysis (PCA), allowing in principle to identify also propagating signals, and therefore investigate both the space and time behavior. Conventional PCA have been employed in Antarctica in the past mostly to reveal the trends only. I believe that the additional information that Complex PCA can provide about the phase changes, can improve the power of analysis of the GRACE signal, and it would be very nice to see a discussion of the new insight that Complex PCA can provide in this specific case over Conventional PCA. However, and unfortunately, the authors do not show that.

**Response:**

We have added literature, description and formulas in the Methods to illustrate the reliability of CPCA, and more detail has been added to the Discussion to improve the interpretation of the results and make the information easier to understand.

A disadvantage of the PCA method is that it can only detect standing waves rather than advancing waves due to the absence of phase information. To overcome this, Horel (1984) introduced phase information into PCA to identify traveling and standing waves (Pfeffer et al., 2010; Kichikawa et al., 2015; Zhan et al., 2017) and named the method complex principal component analysis (CPCA). The CPCA has been widely used in the fields of atmosphere, ocean and earth science to identify information such as advancing waves of observation fields (Harlander et al., 2014; Hoff et al., 2016; Von et al., 2018; Walcker et al., 2015). The CPCA method transforms the original data set and its Hilbert transform into a complex time series and then conducts a principal component analysis by calculating the covariance or complex characteristic vectors of the cross-correlation matrix.

Therefore, the CPCA method still performs PCA analysis, which has the same theoretical basis as the PCA method, except that the input data are the constructed complex data, which provides more implicit information of the data set itself.

Second, like PCA, CPCA does have some potential drawbacks that are mainly related to result interpretation. This is mainly because these methods depend upon the eigen-decomposition of positive semi-definite matrices based on the use of an orthogonal transformation to convert a set of possibly related variables into a set of linearly independent variables and display patterns of similarity of the observations and variables as points in maps. This transformation is defined based on the guideline that the first principal component has the largest possible variance and therefore accounts for as much of the variability in the data as possible. Therefore, the spatial mode has the ability to reveal the patterns of trends or spatial characters corresponding its principal component of mass change, but the magnitude does not represent the actual true trend value, for the observation matrices has been normalized before eigen-decomposition.

Revised include:

**Pg4-5 lines 109-138:**

"Before CPCA, a complex observation sequence should first be constructed using a real observation series. For a time varying observation vector $u_j(t)$, its Fourier expansion is:

$$u_j(\mathrm{t}) = \sum_{\omega} \left[ a_j(\omega)\cos(\omega t) + b_j(\omega)\sin(\omega t) \right]. \tag{1}$$

In the above expansion, $j$ stands for the location of the observation point, $\omega$ is the Fourier frequency and $t$ is the observation time. In order to describe the propagation characteristics of a time series, it is necessary to construct the imaginary part and convert it into a complex observation sequence. The complex observation sequence can be expressed as:

$$U_j(t) = \sum_{\omega} c_j(\omega)e^{-i\omega t}. \tag{2}$$

Here, we define $c_j(\omega) = a_j(\omega) + ib_j(\omega), \mathrm{i} = \sqrt{-1}$. Then the Eq. (2) can be expanded as:

$$\begin{aligned} U_j(t) &= \sum_{\omega} \left[ a_j(\omega)\cos(\omega t) + b_j(\omega)\sin(\omega t) \right] + i\left[ b_j(\omega)\cos(\omega t) - a_j(\omega)\sin(\omega t) \right] \\ &= u_j(t) + i\mathrm{v}_j(t) \end{aligned} \tag{3}$$

The real part of Eq. (3) is the original observation sequence and the imaginary part is the Hilbert

transform of the real part, which does not change the amplitude of each component of $u_j(t)$. However, the phase of each spectral component is advanced by $\pi/2$.

The traditional PCA is the principal component analysis of the real observation vector, whereas the CPCA analysis is the principal component analysis of the complex vector constructed. After the normalization of the complex observation vectors, that is the average value is subtracted from the complex observation vector of each observation point, and then divided by the standard deviation the complex correlation matrix of the observation point can be expressed as:

$$
\begin{bmatrix}
r_{11} & r_{12} & \cdots & r_{1n} \\
r_{21} & r_{22} & \cdots & r_{2n} \\
\vdots & \cdots & \cdots & \vdots \\
r_{n1} & r_{n2} & \cdots & r_{nn}
\end{bmatrix}.
\tag{4}
$$

Here $r_{jk}$ represents the multiple correlation coefficients between the $j$th and $k$th observation points.

CPCA compresses information using the least complex eigenvector $e_{jn}$ of correlation matrix (Eq. 4) and the complex principal component $p_n(t)$, because the correlation matrix (Eq.4) is a Hermitian matrix including n real eigenvalues $\lambda$. $\lambda_j / \sum_{i=1}^{n} \lambda_i$ denotes the contribution percentage of the $j$th principal component.

Observation vector $U_j(t)$ can be expressed as the sum of N principal components,

$$
U_j(t) = \sum_{n=1}^{N} e_{jn}^* p_n(t),
\tag{5}
$$

where * stands for the complex conjugate, and both complex principal components and complex eigenvectors are orthogonal. The $n$th complex eigenvector element $e_{jn}$ can be expressed as

$$
e_{jn} = \left[ U_j(t) * p_n(t) \right]_t = s_{jn} e^{i\theta_{jn}}.
\tag{6}
$$

Where, $e_{jn}$ indicates the multiple correlation relationship between the jth time sequence and nth principal component. $s_{jn}$ and $\theta_{jn}$ are respectively correlative order of magnitude and phase.

$[\cdots]_t$ signifies the average of times. The time sequence elements of principal components can be expressed as the functional form of amplitude $T_n$ and phase $\Phi_n$.

$$P_n(t) = T_n(t)e^{i\Phi_n(t)}. \tag{7}".$$

**Pg8 line 218:**

"The Response of the Antarctic Ice Sheet Mass Balance to the behavior corresponding the first principal component"

**Pg8 lines 220-223:**

" The spatial mode shows where the mass balance is the most sensitive to the change of its corresponding principal component, the phase distribution indicates the source direction of the possible factors that affected mass balance and the length of the arrow reflects the extent to which the mass in this region responds to the variation of these possible factors."

Related Reference were also supplemented:

(1) Harlander, U., Larcher, T., Wright, G. B., Hoff, M., Alexandrov, K. & C. Egbers: Orthogonal decomposition methods to analyze PIV, LDV, and thermography data of thermally driven rotating annulus laboratory experiments. In Modeling Atmospheric and Oceanic Flows (ed. Larcher, T. & Williams, P. D.), 315–336, 2014.

(2) Hoff, M., Harlander, U., & C. Egbers: Experimental survey of linear and nonlinear inertial waves and wave instabilities in a spherical shell. Journal of Fluid Mechanics, 789, 589-616, 2016.

(3) Von Larcher, T., Viazzo, S., Harlander, U., Vincze, M., & A. Randriamampianina: Instabilities and small-scale waves within the Stewartson layers of a thermally driven rotating annulus. Journal of Fluid Mechanics, 841, 380-407, 2018.

(4) Walcker, R., E.J. Anthony, C. Cassou, R.C. Aller, A. Gardel, C. Proisy, J-M Martinez, F. Fromard: Fluctuations in the extent of mangroves driven by multi-decadal changes in North Atlantic waves. Journal of Biogeography, 42, 2209-2219, 2015.

➢ In addition, the analysis of the other climate variables, like El Nino, ENSO, wind and others, is shown in terms of wavelets analysis and the correlation only, that is by its nature only a partial information, and too late in the discussion, while it should be done earlier (in the result session) and the principal component for those variable should be shown too (at least the temporal part).

**Response:**

Based on the suggestion, we have analyzed the climate variables in the early part of the Discussion and supplemented the related principal component for those variable, such as El Niño, ENSO and wind time series. (Pg9 lines 240-248 and Pg21-23 lines 545-565).

➢ The claimed response of the ice sheet to the climate changes is only supported by correlation index between the climate forcing analyzed and the first 2 principal components of the GRACE-derived signal. The author themselves recognize that this is not a proof, and the last line of the conclusion is "To fully understand the causes of changes in ice sheet mass, other principal components need to be analyzed".

**Response:**

We have revised the last line of the conclusion sentence to "To fully understand the causes of changes in ice sheet mass, other phenomena such as enhanced ice-ocean interaction and ice discharge, intrusion of warm saline water on retrograde slopes should be included in the analysis and eventually excluded from the list of possible effects" to make a clearer statement based on the suggestion. (Pg12 lines 361-364).

From the result of CPCA, we can see that the first principal component explains 83.73% of the mass change in the ice sheet, and the second component explains 2.57% of that change in the ice sheet. The contribution of the first two principal components accumulations to the mass anomalies of ice sheet is 86.29%.

Here, we note that the behavior of the first two principal components plays an important role in the mass anomalies of ice sheets and their contribution to the mass anomalies change is estimated at 86.29%. We have revised the last line of the conclusion to make it clear.

➢ I believe, that analyzing more component is not the point, but rather taking into account other possible phenomena as forcing of the ice sheet changes. Therefore the claimed implication "climate forcing -> mass variation" is too strong and it is not supported. And considering that the Complex PCA are not common knowledge at least for some of the potentially interested readers, there should be more explanations on how to read the data, which are not easy understandable for everyone,

otherwise. Referring to a previous paper is not helping to make the present paper more readable.

**Response:**

We have revised the related statement sentence to make it easier to understand. The theoretical formulas of CPCA and more explanations on how to read the data have also been supplemented in the Methods and Discussion sections.

Revised include:

**Pg8 line 218:**

"The Response of the Antarctic Ice Sheet Mass Balance to the behavior corresponding the first principal component".

**Pg8 lines 220-223:**

"The spatial mode shows where the mass balance is the most sensitive to the change of its corresponding principal component, the phase distribution indicates the source direction of the possible factors that affected mass balance and the length of the arrow reflects the extent to which the mass in this region responds to the variation of these possible factors."

**Pg11 line291:**

"5.3 Effect of the second and third principal components on Antarctic Ice Sheet Mass Balance".

➢ Another very weak point of this paper is that the analysis of the GRACE data leads the authors to unreasonable estimates of the mass balance in Antarctica. Not enough details are given to be able to point out what could be the reason for this, but the results are completely outside the realm of the possibility, given the strong consensus emerged in the recent years in the scientific community concerning GRACE (and not only GRACE) mass balance in Antarctica (see Shepherd et al. 2012 and 2018 for example).

**Response:**

We carefully re-examined every step of our estimation of the Antarctic mass balance and compared it with that of Nagler et al. (2017) and Shepherd et al. (2018). We found that the reason for this difference is mainly due to signal leakage errors. When estimating the Antarctic mass balance, the

mask method was not used causing the signals leaking into the ocean to be added to the Antarctic ice sheet mass change (enlarging the area), making the estimation result too large (as shown in Figure 1a). This does not affect the result of CPCA. We have corrected this error using the mask method with the drainage basins boundary data definitions by Zwally et al. (2012). We also used the RL06 data to re-estimate the Antarctic ice sheet balance and the estimated value are consistent with the results of Nagler et al. (2017) and Shepherd et al. (2018). We replaced the previous data set with the RL06 data in the revised manuscript (as shown in Figure 1b).

Another reason is that we used an inappropriate ICE-6G model in Antarctica. Shepherd et al. (2018) compared the effects of multiple GIA models on the mass change of ice sheets in the Antarctic region. The results show that the model risks overestimating mass changes in Antarctica compared to other GIA models. We replaced the ICE-6G model with the latest GIA model of Caron et al. (2018) in the revised manuscript (as shown in Figure 2b). Our related results are presented here:

[Figure]

Figure 1. Mass trend of ice sheet in Antarctic. (a) RL05- ICE-6G model;
(b) RL06- GIA model of Caron et al.(2018) ;

Mass balance in Antarctica of our result:

| Basin | Mass balance | Basin | Mass balance |
| --- | --- | --- | --- |
| AIS1 basin 1 | -3.4 ± 11.6 | AIS14 basin14 | -8.9 ± 5.4 |
| AIS2 basin 2 | -2.0 ± 5.8 | AIS15 basin15 | -3.5 ± 1.0 |
| AIS3 basin 3 | 9.7 ± 14.3 | AIS16 basin16 | -0.5 ± 1.3 |
| AIS4 basin 4 | 7.9 ± 2.9 | AIS17 basin17 | 1.9 ± 14.3 |

| | | | |
|---|---|---|---|
| AIS5 basin 5 | 7.0 ± 1.2 | AIS18 basin18 | 9.6 ± 3.7 |
| AIS6 basin 6 | 16.2 ± 1.8 | AIS19 basin19 | -1.3 ± 5.0 |
| AIS7 basin 7 | 16.8 ± 2.8 | AIS20 basin20 | -37.7 ± 6.7 |
| AIS8 basin 8 | 6.3 ± 1.3 | AIS21 basin21 | -58.4 ± 6.3 |
| AIS9 basin9 | 0.5 ± 1.7 | AIS22 basin22 | -49.3 ± 6.9 |
| AIS10 basin10 | -1.8 ± 7.9 | AIS23 basin23 | -12.5 ± 2.6 |
| AIS11 basin11 | -1.4 ± 3.2 | AIS24 basin24 | -9.7 ± 4.4 |
| AIS12 basin12 | -1.0 ± 4.0 | AIS27 basin27 | -1.5 ± 0.9 |
| AIS13 basin13 | -13.4 ± 2.5 | AIS28NAP(b25-26) | -12.5 ± 2.6 |

Revised include:

The sentence of "For the GIA effect in Antarctica, several models have been released since 2012 ( Whitehouse et al.,2012; Ivins et al.,2013; Peltier et al., 2015; Caron et al.,2018). We choose the latest GIA model (Caron et al., 2018), which consists of normalized spherical harmonic coefficients to degree and order 89, to remove the effect of GIA. The new model used a Bayesian framework with many new GPS time series from 459 sites and 11451 RSL data. Related data are available from the link https://vesl.jpl.nasa.gov/solid-earth/gia/." were added in Pg3 lines 86-90.

The "At last, the method of mask was used to remove the effect of signals leakage" was added in Pg6 line 161.

The related Reference were also supplemented:

(1) Nagler;T., D. Floricioiu, A. Groh, M. Horwath, A. Kusk, A. Muir, J. Wuite. Algorithm Theoretical Basis Document (ATBD) for the Antarctic_Ice_Sheet_cci project of ESA's Climate Change Initiative, version 2.1, 15 December 2017.

(2) Shepherd, A., Ivins, E., Rignot, E., Smith, B., van den Broeke, M., Velicogna, I., Whitehouse, P., Briggs, K., Joughin, I., Krinner, G., et al.: Mass balance of the Antarctic Ice Sheet from 1992 to 2017, Nature, 556, 219–222, doi:10.1038/s41586-018-0179-y, 2018.

➢ The authors devote far too much space to discussing this mass balance (it is also put as the first item in the conclusion), comparing with works that are outdated, and not discussing any of the actual

relevant work on the topic. A weak explanation about their results being quite "different" from the main literature just demote the whole work. Most importantly, it can be true that the CPCA analysis that the author perform on the GRACE data is not affected by this, but the fact that the GRACE-derived mass balance is so "off", raises the doubt that the processing could be correct, but the data to process could be wrong.

**Response:**

Based on your suggestion, we shortened this section in the revised manuscript and added the most recent literature (Pg7-8 lines 200-217).

We have examined all steps of the estimation of the Antarctic mass balance and compared these with Nagler et al. (2017) and Shepherd et al. (2018). We found that the reason for this difference is mainly signal leakage errors. When we estimated the Antarctic mass balance, the mask method was not used, causing the signals leaking into the ocean to be added to the Antarctic ice sheet mass change and making the estimation too large. We have corrected this error using the mask method and the drainage basins boundary data definitions by Zwally et al. (2012).

Specific comments:

In the following I make more detailed comments.

➢ L39-49: Too much. Shorten or remove it. It's not relevant for the paper.

**Response:**

We have Shorten this paragraph in the revised manuscript based on your suggestion (Pg2 lines 37-41).

➢ L52-55: Here GIA, which is not a tectonic process, is missing from the picture.

**Response:**

We added this information in the revised manuscript and the sentence is now "…geological tectonic processes except glacial isostatic adjustment (GIA) deep inside the earth…" (Pg2 line 44).

➢ L62-65: Here the Ocean interaction (which is the most important) is missing from the picture.

**Response:**

We have added this information in the revised manuscript and the sentence is now "The mass

change of the ice sheet is the result of interactions between the ocean, ice sheet and the atmosphere." (Pg2 lines 53-54).

➢ L77: Why RL05 and not RL06? Since there's a new release, the reason of choice of the release should be mentioned.

**Response:**

For technical reasons, there was a lag in the time required to get the latest data release. We now have the latest data and have replaced the RL05 data with the newer RL06 version data (Pg3 line 68).

➢ L80-88: Since the RL06, the description of improvement in RL05 is outdated. I can understand the study has been performed on "old" release, but more words should be spent also on the new release RL06, so not to give the idea that RL05 is the last one or the most up to date.

**Response:**

We have replaced the RL05 data with the newer RL06 version data. A description of the new release RL06 data has been added in the revised manuscript and we removed information about RL05.

The description of the new release RL06 is: "The main improvements in the RL06 products were as follows: (1) a reprocessed GPS constellation consistent with ITRF2014/IGS14; (2) reprocessed RL03 Level-1B products for K-band and star camera instrument data; (3) updated models for ocean tides (FES2014) and Atmosphere and Ocean Dealiasing (AOD1B RL06); (4) modified parameterization of orbit and instruments. The resulting RL06 time series of monthly GRACE Level-2 spherical harmonics with its underlying processing standards will then serve for the continuation with GRACE-FO (Follow-on) data with the idea of harmonizing the two time series. This will ensure the consistency between GRACE and GRACE-FO time series for all scientific studies." (Pg3 lines 71-76).

➢ L92: ICE-6G is inadequate for Antarctica. IJ05-r2 or W12 would be a much better choice. Since 2012 there are several papers about GIA in Antarctica, and several GIA models are available (Caron et al 2018, is one of the most recent and available). I understand that for this work trends like GIA do not matter, but I still don't understand why chosing one of the most inadequate GIA for Antarctica. Probably not using GIA would be a better option, and it would be a good idea to make

a comparison with and without GIA correction: to show that makes no difference.

**Response:**

Based on your suggestion we replaced the ICE-6G model and compared it with, and without, GIA correction. The two results were similar (Fig 2 a. and Fig 2b.).

The related description is replaced with "…For the GIA effect in Antarctica, several models have been released since 2012 ( Whitehouse et al.,2012; Ivins et al.,2013; Peltier et al., 2015; Caron et al.,2018). We choose the latest GIA model (Caron et al., 2018), which consists of normalized spherical harmonic coefficients to degree and order 89, to remove the effect of GIA. The new model used a Bayesian framework with many new GPS time series from 459 sites and 11451 RSL data. Related data are available from the link https://vesl.jpl.nasa.gov/solid-earth/gia/. (Pg3 lines 86-90).

Reference:

(1) Whitehouse, P. L., Bentley, M. J., Milne, G. A., King, M. A. & Thomas, I. D. A new glacial isostatic adjustment model for Antarctica: calibrated and tested using observations of relative sea-level change and present-day uplift rates. Geophys. J. Int. 190, 1464–1482 (2012).

(2) Ivins, E. R. et al. Antarctic contribution to sea level rise observed by GRACE with improved GIA correction. J. Geophys. Res. Solid Earth 118, 3126–3141 (2013).

(3) Caron, L., Ivins, E. R., Larour, E., Adhikari, S., Nilsson, J., & Blewitt, G. (2018). GIA model statistics for GRACE hydrology, cryosphere, and ocean science. Geophysical Research Letters, 45. https://doi.org/10.1002/2017GL076644

[Figure]

Figure 2. Mass trend of ice sheet in Antarctic: (a) Grace result, (b) Grace-GIA of Caron.

➢ L100: Section 3.1 is useless, we already know all this. Use only formulas that are new or serve some

purpose in the paper.

**Response:**

We removed this section based on your suggestion.

➢ L109/L113: I think that the correct terminology is "linearly independent variables" not "uncorrelated", which has a different mathematical/statistical meaning.

**Response:**

We have changed the terminology here and throughout the manuscript.

➢ L135: I don't understand how there could be missing points? Are the missing point in time?

**Response:**

Yes, the missing data here indicates that the time series of mass change at a point shows discontinuous in time. We have added explanation to clarify this.

Some monthly GRACE gravity solutions are not provided due to the data quality itself. Therefore the time series of mass change at a point can show discontinuous and sudden transitions. Before CPCA analysis, the data should be continuous and interpolation of the missing data is needed. To avoid ambiguity, we clarified this requirement in the revised manuscript.

The clarification is "In this way, we obtained the time sequence of mass changes from January 2003 to August 2016 at each grid point. However, some monthly GRACE gravity solutions were not available due to the data quality. Therefore the time series of mass change at one point shows discontinuous and sudden transitions. Before applying the CPCA analysis, we need to interpolate the missing data to make the time sequence continuous. We used a spline function to interpolate missing data in the time series. The rate of change of the ice sheet mass based on GRACE data was estimated at each grid point using least squares to fit a linear trend, an annual signal and a semi-annual signal. At last, the method of mask was used to remove the effect of signals leakage." (Pg6 lines 155-159).

➢ L145: The accuracy of RL05 is lower than RL06. A reader might wonder again why not using RL06.

**Response:**

We have replaced the RL05 data with the new released RL06 version data.

➢ L147: The GIA component is the most inaccurate. Did you take that into account? How? With only one GIA model without errors as the ICE-6G is impossible to have the correct idea of the uncertainty

on GIA.

**Response:**

Yes, you are correct. The GIA component is the most inaccurate. Shepherd et al. (2018) assessed some GIA models and found that the ICE-6G risks overestimating mass changes in the west Antarctic. We have replaced the ICE-6G model with the latest GIA model of Caron et al. (2018).

➢ L147-149: All this stuff if it were correct would be redundand with all the previous work, so it's rather useless to discuss it here. It should at least put in comparison with previous works and with the ESA CCI AMB for example, which is based on published work ([http://esa-icesheets-antarctica-cci.org/index.php?q=GMB](http://esa-icesheets-antarctica-cci.org/index.php?q=GMB) and [https://data1.geo.tu-dresden.de/ais_gmb/](https://data1.geo.tu-dresden.de/ais_gmb/)). The pattern found by the authors is quite the same as fig.1 (from the esa cci amb) but the ampliture are not. B19-B27 mass balance should be about -180 Gt/yr and not 258.5 Gt/yr:

AIS19 basin 19 0.7 ± 3.1

AIS20 basin 20 -35.9 ± 5.6

AIS21 basin 21 -55.0 ± 8.1

AIS22 basin 22 -50.8 ± 9.8

AIS23 basin 23 -9.4 ± 4.9

AIS24 basin 24 -10.7 ± 4.7

AIS27 basin 27 1.0 ± 3.5

AIS28 NAP(b25-26) -18.9 ± 5.9

TOT -179.0 ± 17.2

**Response:**

We carefully examined every step of the manuscript's estimation of the Antarctic mass balance and compared it with that of Nagler et al. (2017) and Shepherd et al. (2018). We found that the reason for this difference is mainly signal leakage errors. When we initially estimated the Antarctic mass balance, the mask method was not used. This caused the signals leaking into the ocean to be added to the Antarctic ice sheet mass change and made the estimation result too large (as shown in Figure 1a). We have corrected this error using the mask method (as

shown in Figure 1b) and the drainage basins boundary data definitions by Zwally et al. (2012).

Another reason for overestimation is our use of an inappropriate ICE-6G model in Antarctica. Shepherd et al. (2018) compared the effects of multiple GIA models on the mass change of ice sheets in the Antarctic region. Their results show that ICE-6G model has the risk of overestimating mass changes in Antarctica compared to other GIA models. We replaced the ICE-6G model with the latest GIA model of Caron et al. (2018) in the revised manuscript.

We have also used the RL06 data to re-estimate the Antarctic ice sheet balance, and the estimated value is consistent with the results of Nagler et al. (2017) and Shepherd et al. (2018). The mass change trend pattern is similar to the result of ESA CCI AMB (https://data1.geo.tu-dresden.de/ais_gmb/). We also added a comparison with previous works. Below is the comparison between ESA CCI AMB and our results:

|  | ESA CCI AMB | Our result |
|---|---|---|
| AIS19 basin 19 | $0.7 \pm 3.1$ | $-1.3 \pm 5.0$ |
| AIS20 basin 20 | $-35.9 \pm 5.6$ | $-37.7 \pm 6.7$ |
| AIS21 basin 21 | $-55.0 \pm 8.1$ | $-58.4 \pm 6.3$ |
| AIS22 basin 22 | $-50.8 \pm 9.8$ | $-49.3 \pm 6.9$ |
| AIS23 basin 23 | $-9.4 \pm 4.9$ | $-12.5 \pm 2.6$ |
| AIS24 basin 24 | $-10.7 \pm 4.7$ | $-9.7 \pm 4.4$ |
| AIS27 basin 27 | $1.0 \pm 3.5$ | $-1.5 \pm 0.9$ |
| AIS28 NAP(b25-26) | $-18.9 \pm 5.9$ | $-12.5 \pm 2.6$ |
| TOT | $-179.0 \pm 17.2$ | $-182.9 \pm 13.4$ |

References:

(1) Nagler, T., D. Floricioiu, A. Groh, M. Horwath, A. Kusk, A. Muir, J. Wuite. Algorithm Theoretical Basis Document (ATBD) for the Antarctic_Ice_Sheet_cci project of ESA's Climate Change Initiative, version 2.1, 15 December 2017.

(2) Shepherd, A., Ivins, E., Rignot, E., Smith, B., van den Broeke, M., Velicogna, I., Whitehouse, P., Briggs, K., Joughin, I., Krinner, G., et al.: Mass balance of the Antarctic Ice Sheet from 1992 to 2017, Nature, 556, 219–222, doi:10.1038/s41586-018-0179-y,

2018.

> L160-169: In this paragraph the use of esperssion "climate change" gives the strong impression that the authors have already drawn their conclusion. But they have yet to demonstrate a clear relationship between climate changes and the ice sheet changes they extract in their analysis. So here and in the rest of the paragraph, I wouldn't call it "climate change" of the fist component. Here just call it "the behavior" of the first component.

**Response:**
We have replaced the words "climate change" with "the behavior."

> L166: I'd like to see also the 3rd component. (instead of figure 1, which is useless)

**Response:**
We have added the 3rd component in the revised manuscript (Pg7 lines 194-195).

> L167-168: So far, the authors have not shown a correlation of the climate change with the behavior of the principal component, so that statement is not true.

**Response:**
Yes, you are correct. We have reworded this statement.

> L172-179: Totally redundant (with text above) and wrong.

**Response:**
We have removed this paragraph.

> L184: This work should be compared with GRACE based work (not with altimetry based work). Altimetry and other techniques have much lower accuracy when it comes to mass trend. The most robust results obtained with GRACE are not in agreement with the mass changes obtained here. See also Shepherd et al. 2018. So the statement is false.

**Response:**
We have revised this statement (Pg8 lines 203-205).

> L184: Velicogna and Wahr (2006)?? This is absolutely outdated. They used a very early release of GRACE, a too short time series and a totally unsuitable ensamble of GIA models. It shouldn't be used as reference, for studies that use later release and much longer time series.

**Response:**

We removed this outdated reference based your suggestion.

➤ L184-189: Totally useless discussion (see also my reason in the above introduction).

**Response:**

Based on your suggestion we have removed this outdated reference.

➤ L197-198: Exactly! And you didn't mention the most relevant papers anyway.

**Response:**

We have supplemented the latest results of relevant papers(Pg8 lines 214-217).

➤ L198: "this difference may be due...". Since Shepherd et al 2012 it is clear that all the GRACE derived data agree very well when all the input ingredients are the same and even if the methods are different. So no more excuses. If the numbers doesn't match it means that the authors used something quite different in their processing and it's most likely wrong. GRACE derived estimated are well consolidated, so at this point there's no much room left for discussion here. So there is only one thing to do, find the error and fix it.

**Response:**

As noted above, we have corrected this error. The error was mainly due to leakage effects of signals.

➤ L208-210: I agree on this but it's not a good excuse for getting the numbers wrong.

**Response:**

Yes, you are correct. We have corrected this error. It was mainly due to the leakage effect of signals.

➤ L212: And here again, the authors put their conclusion before showing solid proof of that. As I said before at this point the authors have not shown the correlation with the global climate change pattern yet.

**Response:**

We removed these sentences.

➤ L213: Section 5.2, this part is important and actually show some correlation with the principal component and the global climate dynamic. This must be moved before the discussion as part of the result! Before claiming that the principal component follows a climate change pattern. Note that I am expert in ice sheet changes and not strictly an expert in global climatology, so this section 5.2 is rather difficult to read for me (and more than half of your readers would be like me). So I will point out where it would be nice to have little more information in the text rather than go on internet digging them out from the literature. Here I also note that the previous section about ice mass changes were exaggerately detailed (and I criticised that), if a reader is

not expert in ice sheet changes I admit that some more details could be useful, yet it should be done taking the state of the art into account correctly, which has not been done by the authors.

**Response:**

We replaced the subtitle "The Response of the Antarctic Ice Sheet Mass Balance to the Change in the El Niño Low-Frequency Periodic Signal" with "The Response of the Antarctic Ice Sheet Mass Balance to the behavior of the first principal component" and added content to this section. The additional detail was added in the Discussion section to increase the interpretation of the results and increase reader comprehension.

➢ L219: ... rephrase please. Maybe you mean: reflects changes in the low frequency with 5 year or longer periodicity?

**Response:**

We have rephrased "the low-frequency signal over a period longer than 5 yr" to "the low-frequency with 5 year and longer periodicity." (Pg8 line 231)

➢ L222: The 8.5 and 6.5 years periodic signal in El Nino makes an indeed interesting correlation, but it's not a proof. Since it's not the main energy, do the authors know where this energy is coming from? What are the phenomena that affects that periodicity in El Nino? Is it possible that it is actually a feedback from the ice sheet itself?

**Response:**

We agree with this comment. In addition to the 8.5 and 6.5 years periodic signal, the El Nino also has other signals with periods of 2–3 years. The energy of 8.5 and 6.5 years periodic signal is slightly weaker than the energy of the 2–3-year periodic signals. We were initially unsure of the reasons why the 8.5 and 6.5 years periodic signal, rather than the 2–3 years periodic signal had the greatest impact on the mass of the ice sheet. A possible explanation is the following: First, the mass change of ice sheet in the Antarctic region obtained from the GRACE time-varying gravity solutions is the change of mass anomaly (Subtracted an average), not the undulation of mass. Our analysis is based on this data set of mass anomaly. One purpose of the analysis was to determine major factors that contribute significantly to this anomalous change.

Second, the first component of mass change in the Antarctic region has a low-frequency with 5 year and longer periodicity, indicating that there is an additional factor (for example the El Nino) with the same periodic characteristics that affects the mass change in this region. This periodic signal of the additional factor affects the balance of the mass change in this region, producing a state of positive or negative balance. The ice mass change of input and output caused by this periodic signal in the factor are not equal. Therefore, this periodic signal becomes the main factor affecting the mass change.

Last, although the 2–3 years periodic signal in El Nino has more energy and also has an impact on the mass of the Antarctic region, the ice mass change of input and output caused by this periodic signal are equal, resulting in an mass change in an equilibrium or weak equilibrium state. Therefore, their contributions to the mass anomaly are smaller and would be considered as minor (e.g. 2nd, 3rd, or smaller) components.

With regard to "What are the factors affecting the El Niño cycle", there is still a lot of controversy on this topic. Some people found that the occurrence of the El Niño event is related to the change of the Earth's rotation speed. When decelerates, the equatorial atmosphere and seawater is able to obtain eastward inertial force that induced by so-called the "brake effect", which weak the equatorial currents and trade winds. The western Pacific warm water flows eastward and the upward trend of the cold water in eastern Pacific is then blocked, then the El Niño phenomenon occurs due to warm water accumulation accompany with the increase of seawater temperature and the rise of sea-level. More detailed information can refer to Zhao J. and Y. Han (2008). In recent years, some peoples have also proposed some new explanations for the El Niño phenomenon, that is, El Niño may be related to seabed earthquakes, changes in seawater salinity, and changes in atmospheric circulation. About this issue, there is still a lot of controversy. I can't provide you more information and it is indeed beyond my scope.

Is it possible that it is actually a feedback from the ice sheet itself? The correlation analysis results may provide an answer. Table 2 shows that the correlation coefficient between the first principal component and the sea surface temperature anomaly time series after a 48-month moving average (the low-frequency with 5 year and longer periodicity) reaches 0.73 and the change of the first principal component of ice sheet lags behind that of the low-frequency with

5 year and longer periodicity in El Niño by a month. Therefore, the temperature of the sea water changes first and then the ice sheet mass changes. So, based on our understanding, this is not a feedback involving the ice sheet itself.

Table 2. Correlation analysis between different factors and principal components based on Monte Carlo hypothesis testing.

|  | Time lag (month) | First principal component | Second principal component | Low frequency signal of meridional wind | 95% confidence level |
|---|---|---|---|---|---|
| El Niño | −9 | 0.24 | - | - | 0.17 |
| Low frequency signal of El Niño | 1 | 0.73 | - | 0.77 | 0.16 |
| Temperature | 1 | - | −0.85 | - | 0.16 |

Reference:

(1) Zhao, J. and Y. Han: The relationship between the interannual variation of Earth's rotation and El Niño events. Pure and Applied Geophysics, 165:1435-1443, 2008.

➢ L223: "Lag correlation analysis", what does it mean?

**Response:**

This is actually cross correlation analysis. We have changed this here and throughout the manuscript.

➢ L225: what the data in the link are meant to be used by the reader?

**Response:**

These are the monthly mean sea surface temperature anomaly data in Niño 1+2 region. We have provided a more detailed link.

➢ L226-227: "...which is much greater..." I understand the meaning but this can be rephrased better.

**Response:**

We have rephrased this here and throughout the manuscript (Pg9 line 239).

➢ L228-229: This is interesting! Not robust at all but very interesting. And could it be the opposite, i.e. the ice changes affecting El Nino? I remember previous studies (mostly posters) about this in the early stages of GRACE, but they have never been published because (with short time series) the correlation was weak. Here it's clearly more visible, but it is not presented in a very convincing way. The study and especially the presentation can be improved.

**Response:**

Thank you for this insight. The mass change of ice sheet in the Antarctic region obtained from the GRACE time-varying gravity solutions is the change of mass anomaly (Subtracted an average). One of the purposes of the CPCA analysis was to find major components that contribute significantly to this anomalous change based on the principle of maximum variance. The wavelet analysis presents the time-frequency character of each major components and then uses the period signals in time-frequency results to find possible correlation factors.

➢ L230-236: Now you can say this (not before as you did prematurely).

**Response:**

We have noticed this point and changed it.

➢ L259: "researchers" -> studies

**Response:**

We have changed it (Pg10 line 280).

➢ L269/L271: The are at least two "was" that should be "is".

**Response:**

We have changed this (Pg9 line 239).

➢ L277: which signals are you referring to?

**Response:**

This sentence has been changed to "The correlation coefficient of the low-frequency signal between the meridional wind field and sea surface temperature anomaly in the equatorial Pacific is 0.77." (Pg9 line 253).

➢ L282: Section 5.3. This discussion is about the second component, which has annual

periodicity as many other phenomena. It's rather meaningless to find a correlation with an annual signal when you could find a good correlation with any other annual signal in the world. So which one is the most relevant in this case? I think a correlation analysis is insufficient here to determine that. So this is much less interesting and even less solid than the discussion about the first component.

**Response:**

Yes, you are right. Cross-correlation analysis mainly reveals causality between signals of different periods. As you noted, signals with the same period can also show a good correlation. However, for the Antarctic ice sheet, the main influence on its mass change is related to the precipitation conditions as well as the ice-ocean interaction. It should include elements such as global climate change, ocean circulation, and temperature in the Antarctic. Only when the conditions change, affecting the input and output of the mass balance and causing it to be in an unbalanced state, will mass anomalies be produced.

The wind field, Southern Annular Mode Index and air temperature also present the annual cycle signal and it is difficult to distinguish which factor is dominant only relying on the correlation coefficients. However the phase distribution information of the second component (arrows in Fig. 2d) provides additional information. The phase distribution (arrows direction) shows that the source of the impact is mainly from the South Pole. This allows us to relate to temperature changes, as the Antarctic Center is the source of cold and high pressure in the region.

Here, we have reduced this section. The phase information section has been supplemented. The relevant conclusions have been edited for accuracy. A map of the third component has also added to this section.

➢ L300: For the same reason above, the inference here is extremely weak.

**Response:**

We have rewritten this section and removed the relevant conclusions statement.

➢ L337-339: This sentence is weird and if the meaning is correct it is really weak. The second component by definition cannot reflect "the overall increase in the mass of ice sheets in

Antarctica". Overall increase that is not happening by the way.

**Response:**

We have rewritten this section and removed the relevant conclusions statement.

➢ L342-347: Same as above. The GRACE derived mass loss are too strong. And even if they were right they are not worth mentioning in the conclusion, since the mass balance is not the aim of the paper. They could be barely used as validation tool... validation that failed in this case.

**Response:**

Yes, the difference mainly comes from the leakage of the ice sheet. We have located the error and corrected it. The related sentence has been removed based your suggestion.

➢ L356: Could the correlation of annual signal be due to other factors? As I said, there are many phenomena in the world with annual periodicity, each of them would correlate very well or even better. So this is not a solid result.

**Response:**

Yes, your statement is correct. The correlation analysis mainly reveals causality between signals of different periods. However, as you noted, any signals with the same period can also be significantly correlated.

The main influences on the mass change of the Antarctic ice sheet are related to the formation of precipitation conditions and other factors. They should include elements such as global climate change, ocean circulation, and temperature in the Antarctic and etc. Only when the conditions change, affecting the input and output of the mass balance and causing it to be in an unbalanced state, will there be mass anomalies.

The wind field, Southern Annular Mode Index and air temperature also have annual cycle signals. It is difficult to distinguish which factor is dominant relying only on correlation coefficients. But the phase distribution information of the second component (arrows in Fig. 2d) provides additional information. The phase distribution (arrows direction) shows that the

source of the impact is mainly from the South Pole. This allows us to relate to temperature changes because the Antarctic Center is the source of cold temperatures and high pressure in the region.

Here, the relevant conclusions have been edited for clarity.

➢ L365: Rather than other components I'd say that other phenomena should be included in the analysis and eventually excluded from the list of possible cause (or effects).

**Response:** We have edited this sentence based on your suggestion.

➢ The figures are of very poor resolution and not easy to read.

**Response:** we have redrawn the figure 1 to make it clearer.

[Figure]

**Figure 1: Trend of ice sheet mass balance in Antarctica after eliminating the GIA effect. Basins 1–27 are the Antarctic drainage divides defined by Zwally et al. (2002). The calculations are based on the difference between the pan-Antarctic SAR mapping of Rignot et al. (2011a) and six different Landsat 8 velocity mappings. Basins 2, 17 and 18 are complimented with differences in 1997 and 2009 SAR velocities poleward of 82.5° S (Scheuchl et al., 2012). The definitions of the West Antarctic ice sheet (Basins 1 and 18–23), the East Antarctic ice sheet (Basins 2–17), and the Antarctic Peninsula (Basins 24–27) allocate the drainage systems according to ice provenance with separation of East and West Antarctica approximately along the Trans-Antarctic Mountains.**

---

## Author Comment (AC2) · 3 Jan 2019

General Comments:

➢ This paper uses complex principal component analysis and wavelet amplitude-period spectrum analysis to examine the main drivers of Antarctic mass change. While the study of the effect of quasi-periodic climate patterns such as the El Nino on Antarctica are important, it is misleading to mostly attribute Antarctic mass balance to such events. Furthermore, the GRACE analysis in the paper suffers from some inadequacies. These issues are explained below. Overall, the paper in its current state is not suitable for publication as it provides misleading conclusions. Significant major revisions are required before potential further review.

The paper assumes that Antarctic changes are caused by low-frequency quasi-periodic climate phenomena and atmospheric circulation patterns, and attempts to divide the attribution of ice sheet mass balance change to such events. However, no evidence is provided for this assumption and yet this is a major underlying assumption of the study. This is misleading, as these events play a much smaller role in the mass balance of the AIS compared to non-cyclic long term patterns such as enhanced ice-ocean interaction and ice discharge, intrusion of warm saline water on retrograde slopes, etc. The assumption of this paper can lead to misleading conclusions as the oscillatory climate events play a much smaller role in the recent mass balance. Yet the authors attribute the components of the CPCA to climate change as far as its effects on periodic climate phenomena such as the El Nino. For example the authors claim "This result shows that changes in the low-frequency signal of the sea surface temperature anomaly in the Niño1+2 region of the equatorial Pacific Ocean may be the main reason affecting the mass change of the ice sheet in Antarctica." Again it is misleading to attribute cyclic SST anomalies from El Nino to the main driver of AIS mass balance change.

   **Response:**

Based on your suggestion, we have added content to improve the manuscript. Supplemental content includes: (1) Literature and words were added in the discussion section to increase the interpretation of the results and to help readers extract information from the results. (2) Discussion sections 5.2 and 5.3 were reorganized and the statements were also revised with more appropriate words; the theoretical formulas of CPCA and additional explanations on how to read the data were also supplemented respectively in the Methods and the Discussion. (3) Conclusion were reorganized and the statements were also revised with more appropriate words. (4) The method of mask was used with the drainage basins boundary data to remove the effect of signals leakage. (5) The related literature are also supplemented in the reference section.

The mass balance of the Antarctic ice sheet, in addition to being affected not only by the quasi-periodic climate phenomena and atmospheric circulation, but also exposed to the influences of non-cyclic long term patterns, such as enhanced ice-ocean interaction and ice discharge, intrusion of warm saline water on retrograde slopes. The expression of these issues in the manuscript may be ambiguous and we have edited them in the revised manuscript.

In recent decades, various techniques have been developed to measure changes in ice-sheet mass, based on satellite observations of their speed (Rignot et al., 2002), volume (Wingham et al., 1998) and gravitational attraction (Velicogna et al., 2006) combined with modelled surface mass balance (Van et al., 2018) and glacial isostatic adjustment. There also have been more than 150 assessments of ice loss from Antarctica based on these approaches since 1989, which provide similar results over the period 1992–2011. Shepherd et al. (2018) extended this assessment to include twice as many studies, doubling the overlap period and extending the record from 1979 to 2017. They also found that there has been a large mass loss in West Antarctica with $159 \pm 26$ billion tons per year and attributed this mass loss to be ocean-driven.

Raphael et al. (2016) studied the Amundsen Sea low (ASL) and found that the Amundsen–Bellingshausen Sea (ABS) region exhibits some of the largest inter-annual atmospheric circulation variability, due in part to orographic forcing and in part to its location in the South Pacific, where atmospheric Rossby waves associated with ENSO variability have a year-round influence. The ENSO plays a significant role in determining the depth of the ASL. The most

energetic Rossby waves associated with ENSO variability in the Southern Hemisphere occur in spring, and hence the strongest correlations between ENSO variability and the ASL generally occur in spring. In its La Niña phase, in spring, ENSO is associated with a deeper ASL and with warm air advection toward the Antarctic Peninsula and West Antarctica. However, from spring to summer the sign of the correlation of the phase of ENSO with respect to air temperature anomalies over Antarctica reverses in many locations. They pointed out that the ASL is an important circulation feature that influences West Antarctic climate variability. The ASL has deepened in recent decades with potential impacts on the regional climate through its influence on the meridional wind field. Some research has suggested that tropical teleconnections have contributed to atmospheric warming in West Antarctica and across the peninsula (Ding et al. 2011; Schneider et al. 2012a), and to sea ice loss in the Bellingshausen Sea (Li et al. 2014). The ASL may be related to the variability of the SAM (e.g., Fogt et al. 2011) and ENSO (e.g., Lachlan-Cope and Connolley 2006). Paolo et al. (2018) pointed out that studies correlating ENSO tropical forcing with Pacific sector climate indicators, such as the Amundsen Sea Low strength, sea-ice extent, and AP temperature, found that correlations with ENSO are significant for some seasons but not for others, with reversals of the sign of the correlation from season to season in some cases. The dominant effect of El Niño on the Amundsen Sea ice-shelf mass is the increased basal melting associated with the onshore flow of Circumpolar Deep Water and coastal upwelling as westerly wind stress intensifies.

Paolo et al. (2018) also pointed how the El Niño /Southern Oscillation affects the height and mass of ice shelves in the Amundsen Sea sector of the West Antarctic Ice Sheet. The response in height is the combined effect of two opposing processes, which are both intensified during El Niño events: surface snow accumulation and ocean-driven basal melting. The result is an overall height increase, but net mass loss, since the ice lost from the base has higher density than the fresh snow being gained at the surface. Ice-shelf response to ENSO variability is strongest between the Dotson and Ross ice shelves, with a weak response in Pine Island Bay, the Bellingshausen Sea and west of the Ross Sea. Given expected increases in total precipitation and frequency of extreme ENSO events as Earth's atmosphere warms, their results imply that interannual variability of ice-shelf height and mass will also increase,

stressing the need to quantify surface accumulation relative to basal melting to project future changes in Antarctic ice shelves.

Therefore, ocean-driven factors do not contradict the climate change and atmospheric circulation. The factors of enhanced ice-ocean interaction such as intrusion of warm saline water on retrograde slopes is also the result of global climate changes, for the interaction of the ice-ocean has existed since the formation of the ice sheet on the edge of the Antarctic. Changes in the global climate have caused interactions such as ice discharge, intrusion of warm saline water on retrograde slopes enhanced.

Principal component analysis (PCA) is based on the idea of using an orthogonal transformation to convert a set of possibly related variables into a set of linearly independent variables, and display patterns of similarity of the observations and variables as points in maps. This transformation is based on the guideline that the first principal component has the largest possible variance and therefore accounts for as much of the variability in the data as possible. Each succeeding component in turn has the highest overall variance possible under the constraint that it is orthogonal to the preceding components. The highest variance refers to the observation data from Antarctica taken as a whole, rather than only a smaller amount of observation data collected along the coast.

With the accumulation of observation data (for example, over 30-50 years or longer), the non-cyclic long term patterns may be a principal component and have the largest possible overall variance of the observed data. Regarding the present GRACE data, the results do not reflect this phenomenon very well.

In this manuscript, PCA was used to obtain the major principal components, which have the largest possible variance of the observed data and therefore account for the maximum amount of variability in the data. The wavelet analysis shows a time-frequency correlation between principal components and various possible affecting factors. The results indicate that there is a strong correlation between changes in the low-frequency signal of the sea surface temperature anomaly in the Niño1+2 region of the equatorial Pacific Ocean and the first component of mass change in Antarctica.

In addition to the long term ocean driven factors, based on GRACE data during 2003-2016, we also found that low-frequency quasi-periodic signals appeared in the first principal component, which could be important factors affecting the mass balance in Antarctic. The expressions in the manuscript may be ambiguous and so we have replaced them in the revised manuscript.

Revised include:

**Pg2 line 56-59:**

"The mass change of the ice sheet is the result of interactions between the ocean, ice sheet and the atmosphere. These interaction are closely related to the changes in air humidity, atmospheric temperatures, atmospheric circulation, and other climatic factors in the Antarctic region. "

**Pg4-5 lines 109-138:**

"Before CPCA, a complex observation sequence should first be constructed using a real observation series. For a time varying observation vector $u_j(t)$, its Fourier expansion is:

$$u_j(\mathrm{t}) = \sum_\omega \left[ a_j(\omega)\cos(\omega t) + b_j(\omega)\sin(\omega t) \right].$$
(1)

In the above expansion, $j$ stands for the location of the observation point, $\omega$ is the Fourier frequency and $t$ is the observation time. In order to describe the propagation characteristics of a time series, it is necessary to construct the imaginary part and convert it into a complex observation sequence. The complex observation sequence can be expressed as:

$$U_j(t) = \sum_\omega c_j(\omega)e^{-i\omega t}.$$
(2)

Here, we define $c_j(\omega) = a_j(\omega) + ib_j(\omega), i = \sqrt{-1}$. Then the Eq. (2) can be expanded as:

$$U_j(t) = \sum_\omega \left[ a_j(\omega)\cos(\omega t) + b_j(\omega)\sin(\omega t) \right] + i\left[ b_j(\omega)\cos(\omega t) - a_j(\omega)\sin(\omega t) \right]$$
$$= u_j(t) + i\mathrm{v}_j(t)$$
(3)

The real part of Eq. (3) is the original observation sequence and the imaginary part is the Hilbert transform of the real part, which does not change the amplitude of each component of $u_j(t)$.

However, the phase of each spectral component is advanced by $\pi / 2$.

The traditional PCA is the principal component analysis of the real observation vector, whereas the CPCA analysis is the principal component analysis of the complex vector constructed. After the normalization of the complex observation vectors, that is the average value is subtracted from the complex observation vector of each observation point, and then divided by the standard deviation the complex correlation matrix of the observation point can be expressed as:

$$
\begin{bmatrix}
r_{11} & r_{12} & \cdots & r_{1n} \\
r_{21} & r_{22} & \cdots & r_{2n} \\
\vdots & \cdots & \cdots & \vdots \\
r_{n1} & r_{n2} & \cdots & r_{nn}
\end{bmatrix}.
\tag{4}
$$

Here $r_{jk}$ represents the multiple correlation coefficients between the $j$th and $k$th observation points.

CPCA compresses information using the least complex eigenvector $e_{jn}$ of correlation matrix (Eq. 4) and the complex principal component $p_n(t)$, because the correlation matrix (Eq.4) is a Hermitian matrix including n real eigenvalues $\lambda$. $\lambda_j / \sum_{i=1}^{n} \lambda_i$ denotes the contribution percentage of the $j$th principal component.

Observation vector $U_j(t)$ can be expressed as the sum of N principal components,

$$
U_j(t) = \sum_{n=1}^{N} e_{jn}^{*} p_n(t),
\tag{5}
$$

where * stands for the complex conjugate, and both complex principal components and complex eigenvectors are orthogonal. The $n$th complex eigenvector element $e_{jn}$ can be expressed as

$$
e_{jn} = \left[ U_j(t) * p_n(t) \right]_t = s_{jn} e^{i\theta_{jn}}.
\tag{6}
$$

Where, $e_{jn}$ indicates the multiple correlation relationship between the jth time sequence and nth principal component. $s_{jn}$ and $\theta_{jn}$ are respectively correlative order of magnitude and phase. $[\cdots]_t$ signifies the average of times. The time sequence elements of principal components can be

expressed as the functional form of amplitude $T_n$ and phase $\Phi_n$.

$$P_n(t) = T_n(t)e^{i\Phi_n(t)}. \tag{7}$$".

**Pg6 line 148:**

"The mass balance of the ice sheet in Antarctica is the result of interactions of many factors such as the ice sheet, atmosphere, ocean and other factors. Ice sheet changes in mass balance are the result of variations in time of specific climate factors represented by different frequency signals as well as long-term non-periodic changes. "

**Pg8 line 218:**

"The Response of the Antarctic Ice Sheet Mass Balance to the behavior corresponding the first principal component"

**Pg8 lines 219-226:**

"Figure 2 shows the time evolution of the principal component, its corresponding spatial mode, and the phase distribution (arrows) of the first three components derived by CPCA. The spatial mode shows where the mass balance is the most sensitive to the change of its corresponding principal component, the phase distribution indicates the source direction of the possible factors that affected mass balance and the length of the arrow reflects the extent to which the mass in this region responds to the variation of these possible factors. From the phase distribution of first principal component (Fig 2b.), we can see that the factors affecting the mass balance mainly come from the direction of the eastern South Pacific. The ice sheets in the AP and West Antarctica (basins B20 to B27), Wilkes Land (basin B13), and Dronning Maud Land and Enderby Land (basins B4 to B8) (Fig. 2b) areas are the most sensitive to the first principal component change."

**Pg9 lines 240-260:**

"The Antarctic Oscillation Index (Antarctic Oscillation, AAO. http://www.cpc.ncep.noaa.gov/products/precip/CWlink/daily_ao_index/history/method.shtml .), Southern Annular Mode (SAM, https://legacy.bas.ac.uk/met/gjma/sam.html), air

temperature in the Antarctic region (ftp://ftp.cdc.noaa.gov/pub/Datasets/ncep.reanalysis/), and the meridian wind speed in the South Pacific region (−80°S− −40°S) at a height of 700 hPa (ftp://ftp.cdc.noaa.gov/pub/Datasets/ncep.reanalysis/pressure/) were also analyzed using wavelet amplitude-period spectrum method to study the possible relations of this low-frequency signal between different data set. The results of their wavelet amplitude-period spectrum are presented in Fig. 4 and Fig. 5. Both the Antarctic Oscillation (Fig. 4d) and the Southern Annular Mode (Fig. 4f) have significant annual and 2-year periodic signals, the amplitude of the 2-year periodic signal is comparable to that of the annual periodic signal. The Antarctic Oscillation Index and the Southern Annular Mode may be greatly affected by the low-frequency signal during a period during 2 years in the change of ENSO and have a smaller correlation with the first component of ice sheet. However, the results of the first two principal component of meridian wind speed (Fig. 5a and 5b) show that the meridian wind in the South Pacific region has 8.5-year and 6.5-year periodic signals. The correlation coefficient of the low-frequency signal between the meridional wind field and sea surface temperature anomaly in the equatorial Pacific is 0.77. These results indicate that changes in the low-frequency signal of the sea surface temperature anomaly in the Niño1+2 region of the equatorial Pacific Ocean may be the possible factors affecting the mass change of the ice sheet in Antarctica. The phase distribution information (arrows in Fig. 2b) also indicates that the factors affecting the mass balance mainly come from the direction of the eastern South Pacific, and it is more likely that the sea surface temperature anomaly causes changes in atmospheric pressure and meridional wind, and conducts its effect (such as changes in atmospheric circulation, precipitation, enhanced ice-ocean interaction, ice discharge, intrusion of warm saline water on retrograde slopes and etc.) to the Antarctic ice sheet, because the change of the first principal component of ice sheet lags behind that of the low-frequency signal of the sea surface temperature anomaly by a month (Table 2).

**Pg10 line291:**

"5.3 Effect of the second and third principal components on Antarctic Ice Sheet Mass Balance"

**Pg10-11 lines 299-331:**

"The phase distribution information (arrows in Fig. 2d) indicates that the factors affecting the mass balance mainly come from the South Pole. This allows us to relate to the temperature changes, as the Antarctic Central area is the source of cold and high pressure air in the region. The wavelet amplitude-period spectrum (Fig. 4h) also shows that the air temperature in the Antarctic region has similar periodic signals.

Figure 2f shows the spatial mode and phase distribution (arrows) of the third components. The phase distribution of the third principal component shows that the factors affecting the mass balance are mainly along the latitude line. The ice sheets in the basins B21−23 and basin B1 areas are the most sensitive to third principal component change. The wavelet amplitude-period spectrum of the third principal component time series (Fig. 3c) shows that the principal component contains significant periodic signals of 8.5 years, 4 years and 5 years. The energy of the 8.5-year periodic signal is the largest, followed by that of the 4-year periodic signal and the 5-year periodic signal, the energy of the signals with period below 2 years is unstable. From the perspective of phase distribution and cycle components, these factors that affect the third principal component may be correlated with the Southern Annular model index ( Fig. 4b and Fig. 4f) as well as the sea surface temperature anomaly in the Niño 1+2 region during this period. The direction of phase in the basins B21−23 is counterclockwise, while in the basin B18 and the northern of basin B1, the phase is in a clockwise direction. This data suggests that the impact factors may come from the disturbance of small scale local atmospheric circulation.

Raphael et al. (2016) studied the Amundsen Sea Low (ASL) and found that the Amundsen–Bellingshausen Sea (ABS) region exhibits large inter-annual atmospheric circulation variability. This is due, in part, to orographic forcing and in part to its location in the South Pacific, where atmospheric Rossby waves associated with ENSO variability have a year-round influence. The ENSO plays a significant role in determining the depth of the ASL. The most energetic Rossby waves associated with ENSO variability in the Southern Hemisphere occur in spring, and hence the strongest correlations between ENSO variability and the ASL generally occur in this season. In its La Niña phase, in spring, ENSO is associated with a deeper ASL and with warm air advection toward the Antarctic Peninsula and West Antarctica. However, from spring to summer the sign of the correlation of the phase of ENSO with respect

to air temperature anomalies over Antarctica reverses in many locations. The ASL is an important circulation feature that influences West Antarctic climate variability. Observations reveal that the ASL has deepened in recent decades with potential impacts on the regional climate through its influence on the meridional wind field. Some studies have suggested that tropical teleconnections have contributed to atmospheric warming in West Antarctica and across the peninsula (Ding et al. 2011; Schneider et al. 2012), and to sea ice loss in the Bellingshausen Sea (Li et al. 2014). The ASL is probably related to the variability of the SAM (e.g., Fogt et al. 2011) and ENSO (e.g., Lachlan-Cope and Connolley 2006). Paolo et al. (2018) noted out that studies correlating ENSO tropical forcing with Pacific sector climate indicators, such as the Amundsen Sea Low strength, sea-ice extent, and AP temperature, found that correlations with ENSO are significant for some seasons but not for others, with reversals of the sign of the correlation from season to season in some cases. The dominant effect of El Niño on the Amundsen Sea ice-shelf mass is the increased basal melting associated with the onshore flow of Circumpolar Deep Water and coastal upwelling as westerly wind stress intensifies. "

**Pg12 lines 354-365:**

"The effect of the specific factor represented by annual periodicity signals on the Antarctic ice sheet mass balance accounts for 2.57% of the total change in the ice sheet mass in Antarctica. The effect of the third component, which contains significant periodic signals of 8.5 years and 4-5 years, on the Antarctic ice sheet mass balance accounts for 1.87% of the total change in the ice sheet mass in Antarctica. The factors represented by the third component may be related to the small scale local atmospheric circulation change Southern Annular model index and westerly wind from the periodicity of signals and phase distribution.

There are many factors that affect the mass balance of the ice sheet in Antarctica. In addition to the factors of long term ocean driven, we also found that the low-frequency quasi-periodic signals appears in the first principal component based on GRACE data during 2003-2016, which is also maybe another important factors affected the mass balance in Antarctic. To fully understand the causes of changes in ice sheet mass, other phenomena such as enhanced ice-ocean interaction and ice discharge, intrusion of warm saline water on retrograde slopes should be included in the analysis and eventually excluded from the list of possible effects."

**The related reference were also supplemented.**

(1) Rignot, E. & Thomas, R. H. Mass balance of polar ice sheets. Science 297,1502–1506, 2002.

(2) Wingham, D. J., Ridout, A. J., Scharroo, R., Arthern, R. J. & Shum, C. K. Antarctic elevation change from 1992 to 1996. Science 282, 456–458, 1998.

(3) Velicogna, I. & Wahr, J. Measurements of time-variable gravity show mass loss in Antarctica. Science 311, 1754–1756, 2006.

(4) van Wessem, J. M. et al. Modelling the climate and surface mass balance of polar ice sheets using RACMO2–part 2: Antarctica (1979–2016). Cryosphere 12, 1479–1498 (2018).

(5) Shepherd, A., Ivins, E., Rignot, E., Smith, B., van den Broeke, M., Velicogna, I., Whitehouse, P., Briggs, K., Joughin, I., Krinner, G., et al.: Mass balance of the Antarctic Ice Sheet from 1992 to 2017, Nature, 556, 219–222, doi:10.1038/s41586-018-0179-y, 2018.

(6) Harlander, U., Larcher, T., Wright, G. B., Hoff, M., Alexandrov, K. & C. Egbers: Orthogonal decomposition methods to analyze PIV, LDV, and thermography data of thermally driven rotating annulus laboratory experiments. In Modeling Atmospheric and Oceanic Flows (ed. Larcher, T. & Williams, P. D.), 315–336, 2014.

(7) Hoff, M., Harlander, U., & C. Egbers: Experimental survey of linear and nonlinear inertial waves and wave instabilities in a spherical shell. Journal of Fluid Mechanics, 789, 589-616, 2016.

(8) Von Larcher, T., Viazzo, S., Harlander, U., Vincze, M., & A. Randriamampianina: Instabilities and small-scale waves within the Stewartson layers of a thermally driven rotating annulus. Journal of Fluid Mechanics, 841, 380-407, 2018.

(9) Walcker, R., E.J. Anthony, C. Cassou, R.C. Aller, A. Gardel, C. Proisy, J-M Martinez, F. Fromard: Fluctuations in the extent of mangroves driven by multi-decadal changes in North Atlantic waves. Journal of Biogeography, 42, 2209-2219, 2015.

(10) Nagler;T., D. Floricioiu, A. Groh, M. Horwath, A. Kusk, A. Muir, J. Wuite. Algorithm Theoretical Basis Document (ATBD) for the Antarctic_Ice_Sheet_cci project of ESA's Climate Change Initiative, version 2.1, 15 December 2017.

(11) Raphael, M.N., et al., 2016. The Amundsen Sea low: variability, change, and impact onAntarctic

climate. B Am Meteorol Soc., 97, 111–121.

(12) Schneider, D. P., C. Deser, and Y. Okumura: An assessment and interpretation of the observed warming of West Antarctica in the austral spring. Climate Dyn., 38, 323–347, 2012.

(13) Li, X., D. M. Holland, E. P. Gerber, and C. Yoo: Impacts of the north and tropical Atlantic Ocean on the Antarctic Peninsula and sea ice. Nature, 505, 538–542, 2014.

(14) Fogt, R. L., D. H. Bromwich, and K. M. Hines: Understanding the SAM influence on the South Pacific ENSO teleconnection. Climate Dyn.,36, 1555–1576, 2011.

(15) Lachlan-Cope, T. A., and W. M. Connolley: Teleconnections between the tropical Pacific and the Amundsen-Bellingshausen Sea: Role of the El Niño/Southern Oscillation. J. Geophys. Res., 111, D23101, 2006.

➢ Also one has to be careful about correlations. The authors admit that the mechanisms of this proposed relationship have to be explored further in future studies, but one has the be careful with correlations between 5+ year low-frequency SST changes associated with the El Nino and Antarctic mass balance. The conclusion that air temperature is the second dominant effect on the mass of AIS is also based on the correlation between the components of quasi-periodic atmospheric circulation patterns and AIS mass balance, and the same issues and assumptions arise here. The claims of the paper should not be generalized outside of the scope of the study.

**Response:**

We want to stress that there is a strong correlation between changes in the low-frequency signal of the sea surface temperature anomaly in the Niño1+2 region of the equatorial Pacific Ocean and the first component of mass change in Antarctica. However, the interaction (such as precipitation change, enhanced ice-ocean interaction, ice discharge and intrusion of warm saline water on retrograde slopes) between the low-frequency signal of the sea surface temperature anomaly in the Niño1+2 region and mass change in Antarctica is unclear and requires additional study. To avoid ambiguity, we removed the sentence "The detailed influential mechanism still needs further study." We have also reorganized the sections 5.2 and 5.3; the related statement sentences were also revised so that the manuscript conclusions are not outside of the study scope.

➢ As noted in the paper, the 1x1 grid does not represent the true GRACE resolution. Given that the

mass change is obtained by simply fitting the time-series for each grid on a smoothed field, it must be noted that the grids are spatially correlated and the trends of nearby basins (particularly small basins) cannot be considered separately from each other. Furthermore, there is amplitude loss in spatial smoothing so the smoothed spatial field is not the optimal way of getting regional estimates. The authors should use a synthetic field to justify their results (compare true vs. retrieved signal) or alternatively use a mascon solution, which is linked in the paper but never used. The overall loss trend of 248.6 Gt/yr seems really high with respect to other estimates.

**Response:**

We agree that the mass changes in grids are spatially correlated and the trends of nearby basins (particularly small basins) cannot be considered separately from each other. Furthermore, the amplitude of the signal will decrease to different degrees when using different filters.

We compared our filter with the classical filter such as Gaussian filter, Correlated-Error Filter and the combined filter (Gaussian with 300 km smoothing + Correlated-Error) in the literature of Zhan et al. (2015). The literature of Zhan et al. (2015) describes how the smoothness priors method works in removing the noise of GRACE data, and compared the results of this filter with that of Gaussian smoother, Correlated-Error filtering and the combined filter (Gaussian smoother + decorrelation filtering) with the "Real signals." The results demonstrate that the smoothness priors method has the advantages of less reduction in amplitude of signals in high latitudes, preserved more details of short-wavelength components in the results and has less signal distortion at low latitudes. The statistical results of the filtered field show that the result of the smoothness priors method is closest to the actual value of the original field in the minimum value, maximum value and the RMS value. Please refer to Figure 1 and Figure 2 and Table 1 listed in Zhan et al. (2015).

Figure 1a is the simulation of the numerical model of mass change trend (as true signal), Figure 1b is the simulation of stripe noise model, and Figure 1c is the synthesis signal of mass change trend of Figure 1a add Figure 1b. Then convert this synthetic field of Figure 1c into normalized spherical harmonic (SH) coefficients, to degree and order 60. We then applied the smoothness priors method (SPM), Gaussian filter, the correlated error filter and the combined filter (Gaussian + the correlated error filter) on the synthesis signal.

Figure 2 shows the filtering results of different filters. They indicate that the smoothness priors method (Figure 2d) produced less reduction in amplitude of signals in high latitudes, preserved more details of short-wavelength components in the result and has less signal distortion in low latitudes.

Table 1 illustrates statistics results of the numerical model of mass change (Figure 1a), and the filtering results of mass change (Figure 2) by applying different filters on the synthetic mass change model. The statistics results of the filtered field shows that the result of the smoothness priors method is closest to the truth value of the original field (Figure 1a) in the minimum value, maximum value and the RMS value.

[Figure]

Figure 1. (a) The numerical model of mass change trend,
(b) the stripe noise model; (c) synthetic model by (a) + (b).

[Figure]

Figure 2. Results by applying different filters on the synthetic model. (a) The Gaussian filter with a smoothing radius of 300km; (b) the correlated error filter; (c) a 300km Gaussian smoothing after the correlated error filter; (d)the SPM filter.

Table 1.The grid statistics results of the numerical mass change trend model and the filtered mass change results by applying different filter on a synthetic mass change model. Unit: cm.

by applying different filter on a synthetic mass change model. Unit: cm.

| Filter | Minimum value (cm) | Maximum value (cm) | Mean (cm) | RMS (cm) |
|---|---|---|---|---|
| Real signal | −11.45 | 13.15 | −0.0396 | 1.448 |
| Gaussian 300 km (A) | −6.78 | 11.06 | 0.0349 | 1.231 |
| De-correlation (B) | −11.53 | 11.69 | −0.0399 | 1.727 |
| A+B | −6.53 | 11.10 | −0.0349 | 1.196 |
| SPM filter | −8.81 | 11.88 | −0.0399 | 1.371 |

Second, we found that the reason for this difference of mass balance is mainly due to signal leakage errors. When estimating the Antarctic mass balance, the mask method was not used causing the signals leaking into the ocean to be added to the Antarctic ice sheet mass change. This enlarged the area and resulted in over estimation. We corrected this error with the mask method by using the drainage basins boundary data definitions by Zwally et al. (2012), and this does not affect the result of CPCA.

➢ The only agreement was with a selected altimetry estimate. The authors claim the discrepancies with previous studies such as Velicogna and Wahr (2006) are partly due to previous releases of GRACE data and signal attenuation due to smoothing. However, more recent results are also in disagreement (such as Velicogna et al 2014), which use newer releases and a mascon approach. Furthermore, scaling factors were calculated for previous studies using synthetic fields to account for signal attenuation. And such approaches such as the spherical cap approach are in close agreement with other mascon solutions such as the JPL or CSR mascons. The authors should also consider the official mascon solutions of the processing centers as a point of comparison. While previous studies that are claimed here to be suffering from signal attenuation due to smoothing looked at scaling and gain factors, this study does not make an attempt on quantifying any attenuation with a synthetic field.

**Response:**

We found that the reason for the mass balance difference is mainly signal leakage errors. When estimating the Antarctic mass balance, the mask method was not used in the manuscript, causing the signals leaking into the ocean to be added to the Antarctic ice sheet mass change (enlarged the area), making the estimation result too large (as shown in Figure 3a). We have corrected this error with the mask method using the drainage basins boundary data definitions by Zwally et al. (2012). We also replaced the previous data set with the RL06 data in the revised manuscript and re-estimate the Antarctic ice sheet balance (as shown in Figure 3b), and the estimated value is consistent with the results of Nagler et al. (2017) and Shepherd et al. (2018). The mass loss in basins B19 to B27 should be −182.9 ± 12.6 Gt/yr. The following is the mass balance trend figure and the table of mass balance of the different drainage basins in Antarctica.

[Figure]

Figure 3. Mass trend of ice sheet in Antarctic. RL05- ICE-6G model (a);
RL06- GIA model of Caron (2018) (b);

Mass balance of different drainage basins in Antarctica of our result:

| Basin | Mass balance | Basin | Mass balance |
|---|---|---|---|
| AIS1 basin 1 | -3.4 ± 11.6 | AIS14 basin14 | -8.9 ± 5.4 |
| AIS2 basin 2 | -2.0 ± 5.8 | AIS15 basin15 | -3.5 ± 1.0 |
| AIS3 basin 3 | 9.7 ± 14.3 | AIS16 basin16 | -0.5 ± 1.3 |
| AIS4 basin 4 | 7.9 ± 2.9 | AIS17 basin17 | 1.9 ± 14.3 |
| AIS5 basin 5 | 7.0 ± 1.2 | AIS18 basin18 | 9.6 ± 3.7 |
| AIS6 basin 6 | 16.2 ± 1.8 | AIS19 basin19 | -1.3 ± 5.0 |
| AIS7 basin 7 | 16.8 ± 2.8 | AIS20 basin20 | -37.7 ± 6.7 |
| AIS8 basin 8 | 6.3 ± 1.3 | AIS21 basin21 | -58.4 ± 6.3 |
| AIS9 basin9 | 0.5 ± 1.7 | AIS22 basin22 | -49.3 ± 6.9 |
| AIS10 basin10 | -1.8 ± 7.9 | AIS23 basin23 | -12.5 ± 2.6 |
| AIS11 basin11 | -1.4 ± 3.2 | AIS24 basin24 | -9.7 ± 4.4 |
| AIS12 basin12 | -1.0 ± 4.0 | AIS27 basin27 | -1.5 ± 0.9 |
| AIS13 basin13 | -13.4 ± 2.5 | AIS28NAP(b25-26) | -12.5 ± 2.6 |

Revised include:

**Pg6 lines 155-159:**

"In this way, we obtained the time sequence of mass changes from January 2003 to August 2016 at each grid point. However, some monthly GRACE gravity solutions were not available due to the data quality. Therefore the time series of mass change at one point shows discontinuous and sudden transitions. Before applying the CPCA analysis, we need to interpolate the missing data to make the time sequence continuous. We used a spline function to interpolate missing data in the time series."

**Pg6 line 161:**

"At last, the method of mask was used to remove the effect of signals leakage."

➢ Analysis is up to September 2015. Is there a reason for this? Even with the presence of accelerometer issues near the end of the mission, at least another year can be added.

**Response:**

We have replaced the old data set with the new released RL06 data, and extending the record to August 2016.

**Specific Comments:**

➢ Line 62: "The mass change of the ice sheet in Antarctica is the result of interactions between the atmospheric vapor and the surface water resources;" this is a misleading statement. While ultimately the mass balance is the result of surface mass balance (including precipitation) and discharge, this is the interaction of many factors such as ice-ocean interaction, bathymetry, etc. Needs to be clarified.

**Response:**

Based on this suggestion, we replaced the "The mass change of the ice sheet in Antarctica is the result of interactions between the atmospheric vapor and the surface water resources;" with "The mass change of the ice sheet is the result of interactions between the ocean, ice sheet and the atmosphere." (**Pg2 lines 53-54)**

➢ Line 72: Mission ended after 15 years.

**Response:**

The sentence was changed to "It successfully operated for 15 years and ended its mission in August 2016." **(Pg3 line 63)**

➢ Equation 1: If solving for surface density on the left hand side, the coefficient should not be divided by the density of water. Refer to equation (14) of Wahr et al (1998). The coefficient also needs to be in surface density units.

**Response:**

This comment is correct. If solving for surface density on the left hand side in Equation 1, the coefficient should not be divided by the density of water. Here the expression of equation (1) is the equivalent water height (EWH). We have removed this section based on the reviewer's suggestion.

➢ Lines 121-125: this assumes all changes in Antarctica are caused by periodic climate events (such as El Nino etc.). This is not necessarily true and is unjustified. There could well be significant long-term non-periodic changes that are the main drivers of change.

**Response:**

We have revised it to read "… changes in mass balance are the result of variations in time of specific climate factors represented by different frequency signals as well as long-term non-periodic changes. Thus, after obtaining the temporal change series of principal components of mass change in Antarctica, the time-frequency information of quasi-periodic signals from the time series required analysis." (**Pg6 lines 141-145)**

**Technical Comments:**

➢ Line 64: sentence is not very clear. And careful with tense. Maybe "the mass change record contains global and local climate change information across time".

**Response:**

The sentence was changed to "The mass change recorded the global and local climate change information varying in time." based on the suggestion (Pg2 line 56).

➢ Line 115: delete "the" in front of phase information.

**Response:**

We have delete it based on the suggestion (Pg4 line 101).

➢ Lines 147,150: change to "basins B19 to B27", and the corresponding numbers in line 150.

**Response:**

We have edited this sentence here and in other parts of the manuscript (Pg7 lines 172-178).

➢ Line 237: "in analyzing the influence of: : :" makes the sentence longwinded and hard to follow. I think that segment can be removed.

**Response:**

We have removed it based on the suggestion (Pg9 line 261).

➢ Line 238: "had a certain impact" is very vague. You need to be more clear as to their conclusion.

**Response:**

The sentence was changed to "… in the AP during a strong El Niño event cause precipitation changes and sea ice accumulation in the region." (Pg9 line 261)

➢ Line 272: Delete "the" before West Antarctica.

**Response:**

We have removed it based on the suggestion and reorganized this section (Pg8-10 lines 219-289).

➢ Line 345: "i.e." is not appropriate here as the acceleration does not follow from the trend magnitude, it is a separate fact (and it would be helpful to also report the acceleration value if this is the case).

**Response:**

We have removed it based on the suggestion and rewrite this section (Pg12 lines 345-352).